

Atmospheric
Chemistry
and Physics

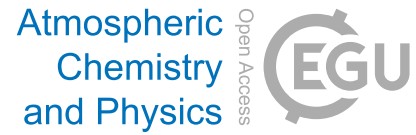

# Canadian and Alaskan wildfire smoke particle properties, their evolution, and controlling factors, from satellite observations

**Katherine T. Junghenn Noyes**[1,2], **Ralph A. Kahn**[3], **James A. Limbacher**[3,4], **and Zhanqing Li**[1,5]

[1]Department of Atmospheric and Oceanic Science, University of Maryland, College Park, MD 20742, USA
[2]Universities Space Research Association, NASA Postdoctoral Program CE1, Columbia, MD 21046, USA
[3]Earth Science Division, NASA Goddard Space Flight Center, Greenbelt, MD 20771, USA
[4]Department of Meteorology and Atmospheric Science, the Pennsylvania State University,
State College, PA 168026, USA
[5]Earth System Science Interdisciplinary Center, College Park, MD 20740, USA

**Correspondence:** Katherine T. Junghenn Noyes (katherine.t.junghenn@nasa.gov)

**Abstract.** TS1 The optical and chemical properties of biomass burning (BB) smoke particles greatly affect the impact that wildfires have on climate and air quality. Previous work has demonstrated some links between smoke properties and factors such as fuel type and meteorology. However, the factors controlling BB particle speciation at emission are not adequately understood nor are the factors driving particle aging during atmospheric transport. As such, modeling wildfire smoke impacts on climate and air quality remains challenging. The potential to provide robust, statistical characterizations of BB particles based on ecosystem type and ambient environmental conditions with remote sensing data is investigated here. Space-based Multi-angle Imaging SpectroRadiometer CE2 (MISR) observations, combined with the MISR Research Aerosol (RA) algorithm and the MISR Interactive Explorer (MINX) tool, are used to retrieve smoke plume aerosol optical depth (AOD) and to provide constraints on plume vertical extent; smoke age; and particle size, shape, light-absorption properties, and absorption spectral dependence. These tools are applied to numerous wildfire plumes in Canada and Alaska, across a range of conditions, to create a regional inventory of BB particle-type temporal and spatial distribution. We then statistically compare these results with satellite measurements of fire radiative power (FRP) and land cover characteristics, as well as short-term climate, meteorological, and drought information from the Modern-Era Retrospective analysis for Research and Applications (MERRA-2) CE3 reanalysis and the North American Drought Monitor. We find statistically significant differences in the retrieved smoke properties based on land cover type, with fires in forests producing the thickest plumes containing the largest, brightest particles and fires in savannas and grasslands exhibiting the opposite. Additionally, the inferred dominant aging mechanisms and the timescales over which they occur vary systematically between land types. This work demonstrates the potential of remote sensing to constrain BB particle properties and the mechanisms governing their evolution over entire ecosystems. It also begins to realize this potential, as a means of improving regional and global climate and air quality modeling in a rapidly changing world.

Please note the remarks at the end of the manuscript.

# 1 Introduction

Wildfires can be significant emitters of trace gases and airborne particles, with the potential to meaningfully impact regional climate conditions as well as short-term local and regional air quality. Smoke emissions alter atmospheric composition by changing the concentration of gases and aerosols across time and space, in turn affecting the surrounding thermal, dynamical, and hydrological conditions. The precise impacts of wildfire emissions depend on a combination of the ambient meteorological and chemical state and, importantly, the composition of the smoke. Although $CO_2$ and water vapor tend to dominate emissions, wildfire smoke includes a rich and complex mixture of many gas and aerosol constituents – most notably the greenhouse gases methane ($CH_4$) and nitrous oxide ($N_2O$), a suite of volatile and semi-volatile organics, light-scattering aerosols and often weakly absorbing soil or dust particles, and the light-absorbing aerosols black carbon (BC) and brown carbon (BrC). Globally, wildfires are the most significant source of light-absorbing airborne particles (Bond et al., 2013; Feng et al., 2013). In addition to exhibiting distinct chemical properties, BC and BrC are optically unique in that BC is highly absorbing across all visible wavelengths, whereas BrC is less absorbing overall and displays enhanced light absorption at shorter wavelengths (Kirchstetter et al., 2004; Samset et al., 2018). In the atmospheric chemistry community, the term "BC" is used to refer specifically to the refractory black carbon component (mid-visible single-scattering albedo, SSA, $\sim 0.4$) that is usually internally mixed within aerosols, derived from in situ light-absorption measurements (Petzold et al., 2013). In contrast, the remote sensing community often uses this term to describe the aerosol types (i.e., entire particles) that exhibit relatively strong (SSA $> \sim 0.7$), spectrally flat light absorption. To avoid confusion, we henceforth refer to these absorbing particles as black smoke (BlS) and brown smoke (BrS), as these terms appropriately describe the spectral dependence of the retrieved SSA without directly connecting to specific chemical constituents. These light-absorbing particles can affect the local radiative budget by warming the ambient air layer and shading the surface, which, in turn, impacts atmospheric stability and may lead to changes in cloud distribution and the water cycle (Albrecht, 1989; Kaufman and Fraser, 1997; Koch and Del Genio, 2010). Smoke aerosols also often contribute towards poor air quality regionally, as particulate matter is dangerous to respiratory health, and the lofting of smoke plumes through plume-rise processes can lead to long-range horizontal transport, so that areas far downwind are also affected. Such plume-rise processes may also lead to smoke aerosols escaping the planetary boundary layer (PBL) and entering the free troposphere (FT), where they can stay aloft for several days or more (Damoah et al., 2004; Taubman et al., 2004; Vant-Hull et al., 2005; Colarco et al., 2004; Kahn et al., 2008; Liu et al., 2014). As such, they have the potential to further impact cloud formation and

lifetime by serving as cloud condensation nuclei (CCN) and possibly increasing cloud albedo via the Twomey effect or, conversely, contributing to droplet warming and evaporation via the semi-direct effect (Kaufman and Fraser, 1997; Koch and Del Genio, 2010; Warner and Twomey, 1967; Hobbs and Radke, 1969; Hansen et al., 1997). The resulting changes in cloud reflectivity and lifetime may then significantly alter climate forcing. Under certain meteorological conditions, smoke plumes can even form pyrocumulonimbus, propelling smoke into the upper troposphere or lower stratosphere (e.g., Peterson et al., 2017); such events are relatively rare and are beyond the scope of the current study, but they are possibly becoming more frequent.

Differences in the optical and microphysical properties of smoke particles indicate that the impacts of wildfires can vary widely. However, the conditions that mediate these differences at the point of emission are not well understood. Wildfires display a range of fire behavior and smoke characteristics that depend on factors such as vegetation type and fuel structure, terrain characteristics, and climate and weather patterns; together, they influence, among other things, the relative degree of flaming or smoldering combustion at the source. Differences in fire regimes and environmental conditions are at least partially linked with differences in smoke particle properties, with evidence suggesting systematic differences in particle size distribution, particle light absorption, and the spectral dependence of absorption (Dubovik et al., 2002; Chen et al., 2008; Eck et al., 2003; Shi et al., 2019; O'Neill et al., 2002). For example, studies have suggested a connection between fire regime and particle size at the point of emission, with smoldering fires (lower combustion efficiency, or CE) generating larger particles than flaming fires (higher CE) under many conditions (Reid and Hobbs, 1998; Reid et al., 2005). These fire regimes have also been linked to smoke particle type – although BC is often the dominant absorbing aerosol component in biomass burning (BB) smoke, smoldering fires tend to produce higher fractions of BrC than flaming ones (Chakrabarty et al., 2010, 2016; Petrenko et al., 2012). Compared with smoldering fires, flaming fires also emit less carbon monoxide (CO), volatile gases, and smoke per unit of fuel consumed (Urbanski, 2013; Wiggins et al., 2021). Both smoldering and flaming regimes occur almost simultaneously in many fires; however, smoldering conditions are more common and may even dominate where fuel is coarse and moist, such as in forests, where the fire can penetrate the organic soil layer. In contrast, flaming conditions dominate over smoldering in regions of fine, grassy fuel that dry out quickly and can produce high-temperature combustion, such as savannas and grasslands (Ottmar, 2001; Urbanski, 2013; Gonzalez-Alonso et al., 2019; van der Werf et al., 2010). These different vegetation types also emit different trace gases when burned, which, in turn, may further impact particle chemistry both near to and downwind of the source (Akagi et al., 2011; van der Werf et al., 2010).

The microphysical properties and mixing state of smoke particles can change dramatically even a short distance away from the source, as aerosols interact with their environment through a variety of complex aging processes. For example, particles may increasingly undergo oxidation as they mix with background air, trace gases, and sunlight, leading to both chemical and structural changes. Particles can also hydrate through the uptake of water vapor, leading to increases in size and light scattering. As smoke cools away from the flame front, semi-volatile gases (known as volatile organic compounds, or VOCs) can condense onto existing emitted particles, creating organic or inorganic coatings that result in increased particle size and alter particle scattering as well as CCN efficiency, especially in the case of BlS (which is hydrophobic in its pure form) (Reid et al., 2005; Zhou et al., 2017; Yokelson et al., 2009; Akagi et al., 2012; Hennigan et al., 2012; Ahern et al., 2019; Dalirian et al., 2018; Kleinman et al., 2020). VOCs can also spontaneously condense into new, very small particles in a process known as secondary organic aerosol (SOA) formation, which results in a higher plume particle number concentration and a smaller average particle diameter across the plume (Akagi et al., 2012; Wang et al., 2013). It is important to note that, in the atmospheric chemistry community, SOA formation is considered to include both new particle formation and VOC condensation onto existing particles. However, from a remote sensing perspective, condensation on existing particles is usually classified as particle growth rather than new particle formation, especially as we often cannot distinguish the condensation of volatile organic gases from hygroscopic growth. Therefore, we consider new particle formation and condensational growth as distinct aging mechanisms here. These and other processes often occur in combinations that may change on relatively short temporal and spatial scales; the factors that determine which mechanism or mechanisms affect the observable particle properties most are currently not well understood.

Based on our current knowledge of the factors controlling smoke particle properties, we might expect that geographic and meteorological conditions are important drivers of particle speciation and plume chemistry. However, to date, there have been no global observational studies to help constrain these relationships on a large scale. As wildfire frequency and severity are expected to increase with global warming, it is becoming increasingly important to improve our understanding of the factors controlling wildfire smoke particle properties. Most current chemical transport and climate models do not discriminate between BlS and BrS, despite their distinct optical and physical properties, which can therefore produce different environmental consequences (Feng et al., 2013; Samset et al., 2018). Models also feature substantial uncertainty about the role that wildfire smoke plays in aerosol–cloud interactions. Thus, better characterization of fire-generated particles is a pressing issue for many modeling efforts. Constraining particle properties and the dominant aging mechanisms in terms of fuel properties and meteorological conditions would contribute greatly towards this goal. Such insight would also have consequences for air quality modeling, as particle speciation and evolution are important factors in determining the atmospheric lifetime of harmful smoke particulates.

Recently developed techniques allow for a better understanding of previously unconstrained wildfire plume heights and particle properties from space, with the potential to characterize wildfire smoke globally by exploring the factors that control emitted and evolved BB particle properties (e.g., Kahn, 2020). These satellite products will achieve their greatest value when applied broadly, to numerous cases over entire ecosystems, yielding statistically robust patterns of smoke-plume behavior. The work presented here takes some first steps toward providing regional constraints on BB particle properties and their dependence on meteorology, vegetation, and burning conditions from space-based observations. Specifically, this study relies on measurements from the Multi-Angle Imaging SpectroRadiometer (MISR) aboard the NASA Earth Observing System's Terra satellite, in conjunction with the MISR Research Aerosol (RA) retrieval algorithm to assess BB particle properties, and the MISR Interactive Explorer (MINX) tool to determine plume height and associated wind vectors, from which aging timescales can be inferred. These methods have been validated in detail against near-coincident in situ observations of smoke plumes from the Biomass Burning Observation Project (BBOP) and the Fire Influence on Regional to Global Environments Experiment – Air Quality (FIREX-AQ) field campaigns (Junghenn Noyes et al., 2020a, b). These experiments demonstrated the strengths and limitations of MISR's ability to (1) constrain particle size, shape, and light-absorption properties, at finer spatial scales and in greater detail than other currently orbiting satellite instruments; (2) map the entire plume, providing context for field observations that are usually only able to observe a disjointed, small percentage of the plume area; and (3) narrow down the likely suite of aging mechanisms acting upon the plume particles at various downwind distances. In both studies cited above, the RA successfully mapped patterns in smoke particle size and light absorption compared to in situ data, with some small exceptions that can be attributed to differences in sampling between the satellite and aircraft and/or the time differences between observations. With the RA, we were also able to infer specific aging mechanisms or burning conditions (e.g., oxidation, secondary particle formation, and gravitational settling), and our results are again supported by the in situ data.

In the current work we apply the MISR tools to a large ensemble of fire plumes across Canada and Alaska that were not constrained by field observations, in order to (1) characterize emitted and evolved smoke particle properties; (2) identify patterns and establish relationships among fuel type, burning conditions, ambient meteorology, and plume properties; and (3) infer the relevant aging mechanisms and

associated timescales from the observed patterns. To this end, we compare the retrieved patterns associated with different ecosystems and environmental conditions with an array of other data, including fire radiative power (FRP) and land cover type from the MODerate resolution Imaging Spectro-radiometer (MODIS), drought severity from the North American Drought Monitor (NADM), and meteorological reanalysis from the Modern Era Retrospective-analysis for Research and Applications (MERRA-2). Trends in particle properties are also studied in the context of smoke age estimates derived from MINX wind vectors. Statistical analysis of the relationships among these observations provides insight into the factors controlling BB particle type emissions and the associated aging processes, directly addressing key elements missing from current climate and air quality modeling efforts. This work is the first installation in a larger effort to constrain BB particle properties globally. Section 2 describes the data and methodology used in this study. Results and discussion are given in Sect. 3. Conclusions and plans for future work are presented in Sect. 4.

## 2 Methodology

### 2.1 The MISR instrument

The MISR instrument is in a polar orbit aboard the NASA Earth Observing System's Terra satellite and has a swath width of $\sim 380$ km. As such, it samples locations at the Equator approximately once every 9 d and locations near the poles every 2 d . MISR offers unique, multi-angle imagery from nine cameras viewing in the forward, nadir, and aft directions along the satellite orbit track, with four spectral bands observed at each angle, centered at approximately 446, 558, 672, and 866 nm (Diner et al., 1998). The use of multiple camera angles makes it possible to retrieve height and motion vectors for clouds and aerosol plumes stereoscopically. This geometrical approach relies on the parallax of contrast features within the plume; therefore, deriving plume height with this method requires that plume features exhibit sufficient optical thickness and contrast relative to the surface. The MISR Interactive Explorer (MINX) software tool (Nelson et al., 2008, 2013) nicely accomplishes these retrievals and was used to derive stereo heights and associated wind vectors for plumes in this work. With MINX, the user manually defines the plume source, plume extent, and wind direction in the MISR imagery, from which MINX retrieves heights and winds locally. MINX has been used in numerous studies, including but not limited to retrieving heights and wind vectors for volcano, wildfire, and dust plumes (Junghenn Noyes et al., 2020a, b; Val Martin et al., 2010, 2018; Scollo et al., 2012; Tosca et al., 2011; Kahn and Limbacher, 2012; Flower and Kahn, 2017a, b, 2018, 2020a, b; Yu et al., 2018; Vernon et al., 2018). Under good retrieval conditions, MINX plume height estimates are accurate within $\pm 0.5$ km or better. In this work, we use the retrieved wind

vectors, along with the distance from the source measured in the images, so that patterns in the evolution of downwind particle properties in smoke plumes can be associated with general timescales. The retrieved stereo heights are used to determine whether the smoke was injected above the planetary boundary layer (PBL), to study how plume height and thickness may inform the evolution of particle size distribution, and to find potential relationships between plume injection and burning intensity, as approximated by fire radiative power (FRP). In addition, the MINX analysis provides some initial insight into the quality of the viewing conditions, as plumes lacking a clear source or easily identifiable wind direction in the satellite imagery can result in low-confidence height retrievals and may need to be excluded from analysis. Lastly, if the MINX plume height is above about 2 km, the MISR images must sometimes be co-registered at the median plume height rather than at ground level to maximize aerosol-type retrieval performance when subsequently using the RA to derive particle properties. For this work, about 50 % of the MINX retrievals were obtained through the MISR Plume Height Project archive (Nelson et al. TS3) (specifically, plumes in 2017 and 2018 from June through August).

Information relating to aerosol type was retrieved using the MISR research aerosol Retrieval Algorithm (RA) (Limbacher and Kahn, 2014, 2019), which compares the multi-angle, multi-spectral MISR observations with simulated top-of-atmosphere (TOA) reflectances to retrieve aerosol optical depth (AOD) and to constrain the particle extinction Ångström exponent (ANG CE5; calculated from measurements at 446 and 866 nm), particle single-scattering albedo (SSA) and its spectral slope, and particle shape (spherical vs. nonspherical) for each $\sim 1.1$ km MISR pixel. The intended use of the RA (e.g., for pollution studies or for wildfire, volcano, or dust plumes) governs the specific set of aerosol components to be included in the algorithm climatology, with each aerosol "type" having a different range of microphysical properties. The particle property information content of MISR observations is qualitative, amounting to three to five size bins (e.g., "small", "medium", and "large"), two to four bins in SSA, and spherical vs. randomly oriented nonspherical particle shapes, under good but not necessarily ideal retrieval conditions (Kahn et al., 2010; Kahn and Gaitley, 2015). For our wildfire studies, the RA includes one nonspherical component (a soil or dust grain optical analogue, based on an optical model derived in Lee et al., 2017) and 16 spherical components ranging in size and SSA values (Table 1 and Table S1 in the Supplement). For light-absorbing aerosols, particle type is further categorized based on the spectral variation in absorption across the visible and near-infrared spectrum, where "flat" aerosols display little to no wavelength dependence and are representative of typical urban pollution or BlS, whereas "steep" aerosols exhibit greater absorption at shorter wavelengths and are more similar to BrS from wildfire smoke (Chen et al., 2008; Sam-

**Table 1.** MISR components from Research Aerosol (RA) retrieval results, using the algorithm version summarized in Sect. 2.1, with a 774-mixture climatology. SSA denotes single-scattering albedo.

| Particle size, shape, and light absorption[a] | $r_e$ (µm)[b] | SSA (446)[c] | SSA (558)[c] | SSA (672)[c] | SSA (866)[c] |
|---|---|---|---|---|---|
| Very small, spherical, and strongly absorbing (flat) | 0.06 | 0.84 | 0.79 | 0.73 | 0.62 |
| Very small, spherical, and strongly absorbing (steep) | 0.06 | 0.76 | 0.80 | 0.83 | 0.76 |
| Very small, spherical, and moderately absorbing (flat) | 0.06 | 0.92 | 0.89 | 0.86 | 0.78 |
| Very small, spherical, and moderately absorbing (steep) | 0.06 | 0.88 | 0.90 | 0.90 | 0.87 |
| Small, spherical, and strongly absorbing (flat) | 0.12 | 0.81 | 0.79 | 0.78 | 0.74 |
| Small, spherical, and strongly absorbing (steep) | 0.12 | 0.72 | 0.80 | 0.87 | 0.84 |
| Small, spherical, and moderately absorbing (flat) | 0.12 | 0.90 | 0.89 | 0.88 | 0.85 |
| Small, spherical, and moderately absorbing (steep) | 0.12 | 0.85 | 0.90 | 0.93 | 0.92 |
| Medium, spherical, and strongly absorbing (flat) | 0.26 | 0.78 | 0.80 | 0.80 | 0.81 |
| Medium, spherical, and strongly absorbing (steep) | 0.26 | 0.70 | 0.80 | 0.88 | 0.89 |
| Medium, spherical, and moderately absorbing (flat) | 0.26 | 0.89 | 0.90 | 0.90 | 0.90 |
| Medium, spherical, and moderately absorbing (steep) | 0.26 | 0.83 | 0.90 | 0.94 | 0.94 |
| Very small, spherical, and non-absorbing | 0.06 | 1.0 | 1.0 | 1.0 | 1.0 |
| Small, spherical, and non-absorbing | 0.12 | 1.0 | 1.0 | 1.0 | 1.0 |
| Medium, spherical, and non-absorbing | 0.26 | 1.0 | 1.0 | 1.0 | 1.0 |
| Large, spherical, and non-absorbing | 1.28 | 1.0 | 1.0 | 1.0 | 1.0 |
| Large, nonspherical, and weakly absorbing | 1.21 | 0.91 | 0.95 | 0.97 | 0.98 |

[a] Particle type includes four elements: size – very small (VSm), small (Sm), medium (Me), and large (La); shape – spherical (Sph) or nonspherical (Nsph); light absorption – non-absorbing (Nab), weakly absorbing (Wab), moderately absorbing (Mab), and strongly absorbing (Sab); spectral light-absorption profile – equal in all spectral bands (flat) or varying between spectral bands (steep). [b] Each component has a lognormal distribution, with a designated effective radius ($r_e$). The minimum and maximum radii as well as the lognormal size distribution width $\sigma$ can be found in Table S1 in the Supplement. [c] Wavelengths are given in nanometers CE4.

set et al., 2018; Limbacher and Kahn, 2014; Andreae and Gelencser, 2006). For each MISR pixel, the RA calculates AOD values for each particle component to create a best-guess mixture representing the aerosol plume composition, such that the simulated TOA reflectances best match those observed in the MISR multi-angle, multi-spectral measurements. This method has already been used for global aerosol typing (Kahn and Gaitley, 2015) and for characterizing particle type in volcanic and wildfire plumes (Toon et al., 2016; Kahn and Limbacher, 2012; Flower and Kahn, 2018, 2020a, b; Junghenn Noyes et al., 2020a, b).

Particle property information derived from MISR and any other passive remote sensing data is based on column-effective, optical measurements rather than on direct sampling. Thus, in this and related work, we refer to the RA measurements of particle size and light absorption as the retrieved effective particle size (REPS; µm) and the retrieved effective particle absorption (REPA; dimensionless), respectively. These terms help reflect both the measured content and the limitations of the retrieved quantities. We use along-plume changes in AOD, REPS, and REPA combined with available meteorological data, MINX stereo heights, and age estimates to help constrain the relevant aging mechanisms for plumes observed under good retrieval conditions. (Retrieved particle property information is reduced when the mid-visible AOD falls below about 0.15 or 0.2, but this is generally not a concern for the well-defined smoke plumes.) For example, decreasing AOD accompanied by de-

creasing REPS downwind might indicate size-selective dilution, whereas uniform particle deposition would feature decreasing AOD accompanied by relatively constant REPS. Similarly, constant AOD accompanied by increasing REPS downwind might signify particle aggregation, whereas constant or increasing AOD accompanied by decreasing REPS could reflect the formation of secondary aerosols. These and other patterns have been observed before with MISR in volcanic and smoke plumes (e.g., Flower and Kahn, 2020a, b; Junghenn Noyes et al., 2020a, b).

The operation of the RA is described by Limbacher and Kahn (2014, 2019). Recently, several advancements have been made to the RA that increase particle property sensitivity, especially for overland retrievals, and are leveraged in this work (see Junghenn Noyes et al., 2020a, b). We have also introduced a revised particle climatology in the RA that is more focused on biomass burning plumes than in previous versions, such as the one used by Flower and Kahn (2017a, b, 2018, 2020a, b). Details of the 17-component optical and physical properties included are given in Table 1 and Table S1 in the Supplement. In general, light-absorbing particles are classified as either strongly light absorbing (mid-visible SSA $\sim 0.80$), moderately light absorbing (SSA $\sim 0.90$), or weakly light absorbing (SSA $\sim 0.95$). Particle size is classified as either "very small" (effective radius $r_e \sim 0.06$ µm), "small" ($r_e \sim 0.12$ µm), "medium" ($r_e \sim 0.26$ µm), or "large" ($r_e > \sim 1.2$ µm), where particles are assumed to be lognormally distributed. The uncertainty in the retrieved AOD

(which can also affect constraints on the particle properties) becomes large once AOD exceeds about 7, as the surface is no longer visible to the MISR; thus, for this work, we only consider RA results in pixels with AOD at or below this threshold.

## 2.2 Experiment setting and case selection

Suitable fires within the ∼ 380 km MISR swath were identified from True Color CE6 imagery and thermal anomalies in coincident MODIS/Terra data using the NASA Worldview web application. The MODerate resolution Imaging Spectroradiometer (MODIS) instrument has a cross-track swath width of 2330 km that provides global coverage every 1 to 2 d. Well-defined plumes of sufficient optical thickness, having a clear source and minimal cloud contamination, were favored for analysis. A total of 663 plumes, burning between 1 May and 30 September and spanning the 4 years of this study (2016–2019), were analyzed. Table 2 quantifies the relative distribution of the observed plumes across year, month, regional location, and land cover type (the latter is discussed more in Sect. 2.3). For a small number of cases, fires in the continental US were included in the study if they were part of a larger complex that burned mostly within Canada, and they are classified here as belonging to the nearest Canadian province.

As Terra crosses the Equator at ∼ 10:30 LT (local time) CE8, the fires considered in this study are restricted to late-morning burns. Although burning usually peaks in the late afternoon, MISR observes a significant number of large, intense plumes, as has been shown in multiple studies previously (e.g., Val Martin et al., 2010; Gonzalez-Alonso et al., 2019) and by the success of the MISR Plume Height Project (Nelson et al. TS4).

## 2.3 MODIS fire radiative power and land cover type

The MODIS/Terra Thermal Anomalies/Fire (MOD14) product was used to identify the fire pixel location and the 5 min FRP values at the time of MISR observation (Giglio and Justice, 2015). Each plume was assigned a mutually exclusive set of hotspots based, first, on which ones fell inside the user-defined MINX boundary and, second, on the proximity to the boundary based on MODIS/Terra RGB imagery from NASA Worldview (https://worldview.earthdata.nasa.gov TS5). Pixels identified with 0 % confidence in the FRP product were ignored, except in cases where a plume did not contain any fire pixels with higher confidence, as these at least provide the locations of the burn and, therefore, land cover type (described below). For all but one plume, at least one fire pixel was detected. The MOD14 product has a spatial resolution of 1 km, and it reports FRP based on a detection algorithm that evaluates differences in the hotspot vs. background brightness temperature using the 4 and 11 µm channels (Giglio et al., 2003). FRP is often used as a qualitative indicator of fire

intensity; however, MODIS may underestimate FRP values under cloudy CE9 or dense-smoke conditions, when the active fire only partly fills the MODIS pixel, and for plumes in the smoldering phase that can exhibit lower radiant emissivity and, therefore, lower FRP values (Kahn et al., 2008).

We systematically associated the fire pixels with annual 0.5 km land cover type data from the MCD12Q1 product (Friedl and Sulla-Menashe, 2019), which includes data from the MODIS instruments on both the Terra and Aqua satellites. We used this information to classify the type(s) of vegetation burning in each hotspot using (a) the International Geosphere–Biosphere Programme (IGBP) classification, which generally categorizes vegetation types based on canopy height, percent cover, woody vs. herbaceous, and evergreen vs. deciduous, and (b) the FAO-Land Cover Classification System (LCCS) surface hydrology layer classification, which provides less specific information but contains several additional categories compared with IGBP. It is important to note that the MODIS land cover type products do not contain sufficient detail to determine the actual *fuel* type consumed by fires, which also depends on a variety of other factors (e.g., meteorology, preexisting burned area, and seasonality). However, land cover and fuel type are highly correlated, and we can use the MODIS product to make inferences as to the types of fuels that are present. (Future work will involve the use of more detailed fuel type information, as discussed in Sect. 4.)

Descriptions of the IGBP land cover types identified in this study are included in Table 3, and descriptions of all land cover types from both products can be found in Tables S2 and S3 in the Supplement. As the MCD12Q1 spatial resolution is finer than that of MOD14, some MODIS hotspots cover multiple land cover types, in which case we assigned land type as a split between the two that comprise the largest fractions of the fire pixel.

## 2.4 MERRA-2 reanalysis

For each plume, we obtain the estimated height of the planetary boundary layer (PBLH) from the MERRA-2 reanalysis model (Bosilovich et al., 2016; Gelaro et al., 2017). The PBLH data are provided at a 0.625° longitude × 0.5° latitude spatial resolution and an hourly temporal resolution, so we choose the data point closest to the time and location of each fire plume origin. Note that, throughout this work, we alternate between using the phrases "above the PBL" and "in the free troposphere (FT)" for plumes that we estimate were injected above the MERRA-2-defined boundary layer. These terms have an identical meaning; the latter is generally used when the emphasis is on smoke transport rather than plume rise.

We calculate atmospheric stability profiles for the column above each plume using three-dimensional (3D) MERRA-2 meteorological data, reported every 6 h at a 0.625° longitude × 0.5° latitude spatial resolution. We consider atmo-

**Table 2.** Distributions of plume number, plume height, boundary layer height, location of burn, and dominant MODIS fuel type shown in three different ways: **(a)** annually, **(b)** monthly, and **(c)** by fuel type. Note that an individual fire may burn in several biomes, so plumes in Table 2c are not in mutually exclusive categories. PBL denotes planetary boundary layer. See footnotes for land type and region/territory abbreviations.

**(a) Plumes by year**

|  | 2016 | 2017 | 2018 | 2019 | Total |
|---|---|---|---|---|---|
| No. plumes | 71 | 319 | 114 | 159 | 663 |
| Median plume height (km)[1] | 1.44 | 1.25 | 1.06 | 1.19 | 1.22 |
| Max plume height (km)[2] | 2.26 | 1.97 | 1.84 | 2.21 | 2.07 |
| No. (%) above 2 km[3] | 13 (18.3) | 35 (10.9) | 6 (5.26) | 15 (9.43) | 69 (10.4) |
| No. (%) above the PBL[4] | 24 (33.8) | 68 (21.3) | 17 (14.9) | 34 (21.38) | 143 (21.57) |
| Median PBL height[5] | 1.69 | 1.61 | 1.52 | 1.47 | 1.57 |
| Dominant land types burned[6] | W. Sav., Sav. | W. Sav., Sav. | W. Sav., evergreen | W. Sav., Sav. | W. Sav., Sav. |
| Dominant regions[7] | Sask. (28 %) | NWT (39 %) | BC (70 %) | AK (47 %) | BC (25 %) |
|  | AK (26 %) | BC (24 %) | Yuk. (9 %) | Alb. (25 %) | NWT (21 %) |

**(b) Plumes by month, aggregated over 4 years**

|  | May | June | July | August | September |
|---|---|---|---|---|---|
| No. plumes | 39 | 51 | 259 | 264 | 50 |
| Median plume height (km)[1] | 1.39 | 1.56 | 1.29 | 1.16 | 0.739 |
| Max plume height (km)[2] | 2.21 | 2.69 | 2.09 | 1.89 | 1.71 |
| No. (%) above 2 km[3] | 7 (17.9) | 11 (21.6) | 27 (10.4) | 24 (9.09) | 0 (0.00) |
| No. (%) above the PBL[4] | 17 (43.59) | 18 (35.29) | 51 (19.69) | 51 (19.32) | 6 (12.00) |
| Median PBL height[5] | 1.44 | 1.71 | 1.63 | 1.54 | 1.07 |
| Median FRP (W m$^{-2}$)[8] | 54.13 | 49.09 | 46.39 | 49.38 | 37.97 |
| Median PBL-top stability (K km$^{-1}$)[9] | 2.61 | 3.95 | 4.43 | 4.33 | 6.51 |
| Dominant land types burned[6] | W. Sav., M.F. | W. Sav., Sav. | Sav., W. Sav. | W. Sav., Sav. | W. Sav., evergreen |
| Dominant regions[7] | Alb. (54 %) | AK (47 %) | AK (34 %) | BC (48 %) | Sask. (42 %) |
|  | Ont. (21 %) | Yuk. (14 %) | NWT (22 %) | NWT (31 %) | BC (36 %) |

**(c) Plumes by land type, aggregated over 4 years**

|  | Evergreen | Mixed forests | Deciduous | Woody savanna |
|---|---|---|---|---|
| No. plumes | 205 | 39 | 3 | 459 |
| Median plume height (km)[1] | 1.19 | 1.22 | 0.584 | 1.23 |
| Max plume height (km)[2] | 2.11 | 2.04 | 1.71 | 2.07 |
| No. (%) above 2 km[3] | 24 (11.7) | 5 (12.8) | 0 (0.0) | 53 (11.5) |
| No. (%) above the PBL[4] | 42 (20.5) | 8 (20.5) | 0 (0.0) | 98 (21.4) |

|  | Savanna | Grassland | Shrubland | Wetlands |
|---|---|---|---|---|
| No. plumes | 312 | 80 | 62 | 2 |
| Median plume height (km)[1] | 1.30 | 1.23 | 1.17 | 1.52 |
| Max plume height (km)[2] | 2.13 | 2.22 | 2.06 | 2.07 |
| No. (%) > 2 km[3] | 39 (12.5) | 8 (10.0) | 4 (6.45) | 0 (0.0) |
| No. (%) above the PBL[4] | 82 (26.28) | 20 (25.0) | 13 (21.0) | 1 (50.0) |

[1] MINX-derived median plume heights above ground level (a.g.l.), averaged across the given category. [2] MINX-derived maximum plume heights above ground level, averaged across the given category. [3] Median plume height must be > 2.0 km a.g.l. [4] Median plume height must be > PBL height + 100 m. [5] Height above ground level, from MERRA-2 reanalysis data. [6] M.F. – mixed forests, Sav. – savannas, and W. Sav. – woody savannas. [7] AK – Alaska, Alb. – Alberta, BC – British Columbia, NWT – Northwest Territories, Ont. – Ontario, and Sask. – Saskatchewan. [8] The median plume MODIS fire radiative power, averaged across the given category. [9] The estimated atmospheric stability at the top of the planetary boundary layer, from MERRA-2 reanalysis data.

https://doi.org/10.5194/acp-22-1-2022 Atmos. Chem. Phys., 22, 1–24, 2022

**Table 3.** Definitions of the land cover types detected in this study, from the MODIS International Geosphere–Biosphere Programme (IGBP) classification method.

| Land cover type | Description |
| --- | --- |
| Evergreen needleleaf forests | Dominated by evergreen conifer trees (canopy > 2 m); tree cover > 60 % |
| Evergreen broadleaf forests | Dominated by evergreen broadleaf and palmate trees (canopy > 2 m); tree cover > 60 % |
| Deciduous needleleaf forests | Dominated by deciduous needleleaf (larch) trees (canopy > 2 m); tree cover > 60 % |
| Deciduous broadleaf forests | Dominated by deciduous broadleaf trees (canopy > 2 m); tree cover > 60 % |
| Mixed forests | Dominated by neither the deciduous nor evergreen (40 %–60 % of each) tree type (canopy > 2 m); tree cover > 60 % |
| Closed shrublands | Dominated by woody perennials (1–2 m height); > 60 % cover |
| Open shrublands | Dominated by woody perennials (1–2 m height); 10 %–60 % cover |
| Woody savannas | Tree cover 30 %–60 % (canopy > 2 m) |
| Savannas | Tree cover 10 %–30 % (canopy > 2 m) |
| Grasslands | Dominated by herbaceous annuals (< 2 m) |
| Permanent wetlands | Permanently inundated lands with 30 %–60 % water cover and > 10 % vegetated cover |

spheric stability as the vertical gradient of potential temperature (Eq. 1) (Holton, 1992 TS6), where $S$ is the stability value at the midpoint of two model levels, $d\theta$ is the calculated difference in potential temperature between the levels, and $dz$ is the difference in geopotential height. Potential temperature is calculated using Eq. (2) (Holton, 1992 TS7), where $T$ and $P$ are the atmospheric temperature and pressure, respectively, at altitude $z$. $P_o$ is the surface pressure (taken as 1000 mbar CE10), $R$ is the gas constant for dry air, and $C_p$ is the specific heat for dry air. For each plume, the temperature and pressure fields were interpolated to the time of MISR observation at the MERRA-2 point closest to the fire location. The height of the stable layer was defined as the height of the first maximum in the stability profile, so long as the stability is at least 1 K km$^{-1}$ TS8 larger than the layers above and below.

$$S = \frac{d\theta}{dz} \tag{1}$$

$$\theta = T \left( \frac{P_o}{P} \right)^{R/C_p} \tag{2}$$

### 2.5   North American Drought Monitor

To evaluate the potential impacts of drought on smoke plume heights and particle properties, we leverage information on drought severity from the Canadian Drought Monitor (CDM) for plumes in Canada and from the United States Drought Monitor (USDM) for plumes in Alaska or just south of the US–Canada border. Both the CDM and USDM are part of the North American Drought Monitor (NADM) effort, a cooperative project between Canada, the US, and Mexico that works to continually monitor drought extent and severity

(Lawrimore et al., 2002) based on the methodology of the USDM (Svoboda et al., 2002). This system uses a blend of drought indicators such as the normalized difference vegetation index (NDVI), streamflow values, the Palmer drought index, and others used by the agriculture, forest, and water management sectors (Agriculture and Agri-food Canada TS9). The synthesis of these reports is analyzed by federal, state, and local academic scientists until a consensus is reached on the best representation of current drought conditions. Assessing drought in this blended manner may be preferable to using just one indicator, as different drought indices measure drought in different ways, and no single index works under all circumstances (Heim, 2002). The NADM index is based on a convergence of evidence from a wide variety of objective inputs and subjective adjustments based on local impacts.

The NADM drought classes range from D0 to D4: D1 to D4 indicate moderate to exceptional drought, and D0 represents abnormally dry conditions. The D0 class is not technically a drought classification, but it might indicate if an area is vulnerable to or recovering from drought. Areas without an assigned drought class are considered to experience normal or wetter-than-normal conditions. The drought categories are based on the percent chance of those conditions occurring over a 100-year period and are classified as follows:

- D0 (abnormally dry) – represents an event that occurs once every 3–5 years;

- D1 (moderate drought) – represents at event that occurs every 5–10 years;

- D2 (severe drought) – represents an event that occurs every 10–20 years;

Atmos. Chem. Phys., 22, 1–24, 2022                    https://doi.org/10.5194/acp-22-1-2022

– D3 (extreme drought) – represents an event that occurs every 20–25 years;

– D4 (exceptional drought) – represents an event that occurs every 50 years (Agriculture and Agri-food Canada TS10).

The USDM is a collaborative effort between the National Drought Mitigation Center (NDMC), the U.S. Department of Agriculture (USDA), and the National Oceanic and Atmospheric Administration (NOAA). It reports the state of drought on a weekly basis and can be accessed at https://droughtmonitor.unl.edu/TS11. The CDM is developed by the National Agroclimate Information Service (NAIS) of the Department of Agriculture and Agri-food (AAFC) and reports on a monthly basis. The data can be accessed at https://open.canada.ca/data/en/dataset/292646cd-619f-4200-afb1-8b2c52f984a2 TS12.

## 3 Results and discussion

Figure 1 maps the plumes used this study over Canada and Alaska, superposed CE11 on the 2017 MODIS IGBP land cover types, and Table 2 shows how these plume observations are distributed by year, month, and MODIS land cover type. The largest number of plumes in our study set was recorded in 2017 (48 % of the total), whereas the smallest number was recorded in 2016 (11 %). Plumes were observed mostly in British Columbia, the Northwest Territories, and Alaska, although a significant number of fires occurred in other provinces and territories. (No suitable plumes were found in Nunavut or east of Ontario.) Most plumes were observed in July and August (79 %), at the peak of the burning season, and during abnormally dry or drought conditions (65 %). Woody savanna was the most common land cover type, followed closely by savanna and evergreen forest. A smaller, although still significant, number of plumes were from fires that at least partially burned in grassland, mixed forest, and open shrubland. Table 3 provides definitions for these and several other land types detected in this study.

We combine the MODIS land cover types into three broad categories to classify the observed fires: (1) "forest" fires (denoted with an "F" where appropriate), which contain any number of MODIS hotspots located in evergreen, deciduous, or mixed forests; (2) "woody" fires (W), which do not burn in forest but have at least 30 % of their hotspots located in woody savanna and up to 70 % located in savanna, grassland, or shrubland; and (3) "grassy" fires (G), which also have no forest hotspots but have at least 70 % of their hotspots in savannas, grasslands, or shrublands, and no more than 30 % of their hotspots in woody savanna. This categorization captures the range of fuel sizes generally included in most classification models (e.g., Ottmar et al., 2001; Scott and Burgan, 2005). Forest plumes are the most likely to contain coarse, woody fuels that tend to maintain their moisture over longer periods of time and often tend to burn in the smoldering phase, whereas grassy plumes are likely dominated by fine fuels that dry out quickly and burn mostly in the flaming phase (Ottmar, 2001; Urbanski, 2013). The woody plume category represents an intermediate step between these two.

There is important seasonal variability in the observed fire types, with G plumes comprising nearly half of all those observed in July and almost none in the colder months of May and September (Fig. 2c). In contrast, despite the overall lower plume numbers in these months, F plumes are dominant in May and September. This dichotomy can be traced to the latitude of the burns – 87 % of G plumes were observed in the Northwest Territories, Alaska, or the Yukon, the three northernmost areas of study, dominated largely by grassy fuels (particularly savannas). Among F plumes, 64 % burned to the south in either British Columbia or Alberta, dominated by forests (Fig. 1). The delay of the peak fire season in the colder regions limits G plume occurrence until later in the summer or even early fall.

Below, we present our analysis of the significant trends in plume heights, particle properties, FRP, and atmospheric conditions, with a focus on the difference between the three broad fire types where possible. Table 4 provides a statistical summary of some of the main smoke plume parameters for each fire type, which are explored in more detail in subsequent figures. Where appropriate, we perform independent $t$ tests to assess if observed differences between fire types are statistically significant ($p$ value less than 0.05). We find that, in most cases, the differences are significant at least between two of the three fire types. This suggests that, although some differences are distinct, many patterns exist on a continuum, and results are likely somewhat dependent on how we define our fire types. For example, plumes in the F category may actually contain only a small percentage of forest. Likewise, the tree cover fraction varies widely within IGBP categories, and satellite data do not reflect the percent of woody/grass fuel actually consumed in a fire. This is why obtaining a large, statistical sampling CE12 is so important. Despite these caveats, many of the patterns that we observe are consistent with current knowledge about fire properties, and the incorporation of additional datasets allows us to build upon this knowledge. The MISR-retrieved particle properties allow us to infer some of the processes affecting particle emissions and evolution, especially when placed in the context of the land cover, FRP, drought level, and plume heights.

### 3.1 Plume height, atmospheric structure, and fire radiative power observations

Plume heights vary considerably across the dataset, ranging from less than 0.2 km to just above 4.0 km a.g.l. (above ground level TS13). In general, however, median plume heights are centered ∼ 1.2 km, and maximum heights are centered ∼ 2.0 km, with only ∼ 10 % of plumes having median heights above 2 km. There is little month-to-month vari-

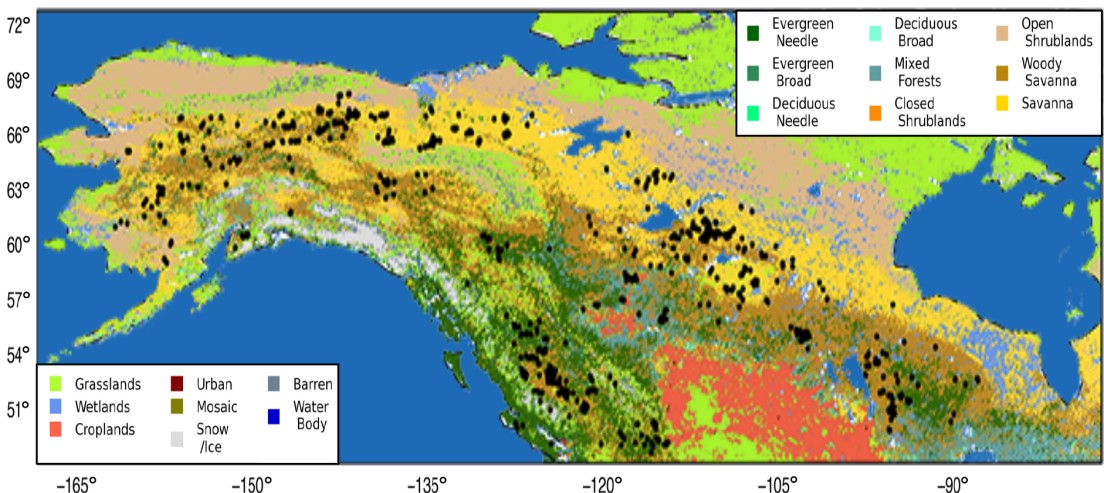

**Figure 1.** Map of all plumes used in this study (black dots) overlaid on the 2017 MODIS IGBP land cover types.

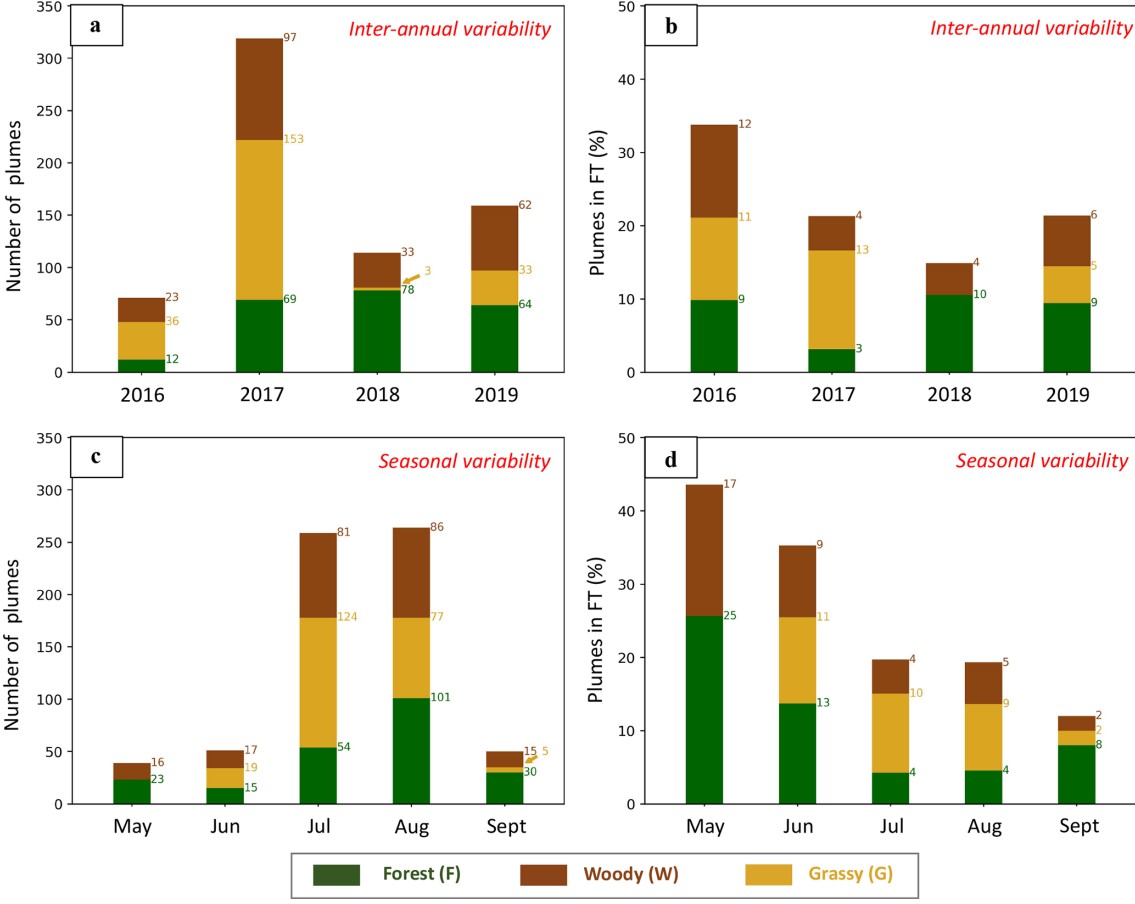

**Figure 2.** Seasonal and inter-annual variability in the plume number **(a, c)** and the percentage of those plumes in the free troposphere (FT) **(b, d)**. Each bar is divided by color according to the relative contribution from each of the three fire types in the given month or year, with quantitative annotations. For example, 39 plumes were identified in the month of May **(c)**, 16 of which were W fires and 23 of which were F fires. Of the 39 fires that month, ∼ 42 % were in the FT (∼ 17 % classified at W fires, and ∼ 25 % classified as F fires) **(d)**. Note that differences in rounding (for the sake of simpler visualization here) produce slightly different numbers compared with the percentage of plumes in the FT given in Table 2. A plume is considered to be in the FT if its median height is 100 m greater than the PBL height as defined in the MERRA-2 dataset.

**Table 4.** Statistical summary of the main smoke plume parameters for each fire category. The following abbreviations are used in the table: FRP – fire radiative power, AOD – aerosol optical depth, ANG – Ångström exponent, SSA – single-scattering albedo, BlS – black smoke, and BrS – brown smoke.

| | Forest fires (F) | | Woody fires (W) | | Grassy fires (G) | |
|---|---|---|---|---|---|---|
| Median plume MODIS | mean: | 40.53 | mean: | 55.19 | mean: | 43.93 |
| per-pixel FRP (W m$^{-2}$) | $\pm\sigma$: | $\pm33.93$ | $\pm\sigma$: | $\pm59.81$ | $\pm\sigma$: | $\pm43.38$ |
| | min: | 6.0 | min: | 4.5 | min: | 0.0 |
| | max: | 229.6 | max: | 367.0 | max: | 306.5 |
| Cumulative plume MODIS | mean: | 1575.5 | mean: | 829.86 | mean: | 682.67 |
| FRP (W m$^{-2}$) | $\pm\sigma$: | $\pm4115.4$ | $\pm\sigma$: | $\pm1805.1$ | $\pm\sigma$: | $\pm1295.1$ |
| | min: | 6 | min: | 4.5 | min: | 0 |
| | max: | 40 254.6 | max: | 20 070.4 | max: | 11 990.7 |
| MISR AOD (558 nm) | mean: | 1.544 | mean: | 1.450 | mean: | 1.147 |
| | $\pm\sigma$: | $\pm1.025$ | $\pm\sigma$: | $\pm1.039$ | $\pm\sigma$: | $\pm0.7877$ |
| | min: | 0.2012 | min: | 0.2239 | min: | 0.1780 |
| | max: | 6.449 | max: | 6.270 | max: | 5.697 |
| MISR ANG (558 nm) | mean: | 1.65 | mean: | 1.70 | mean: | 1.75 |
| | $\pm\sigma$: | $\pm0.278$ | $\pm\sigma$: | $\pm0.252$ | $\pm\sigma$: | $\pm0.243$ |
| | min: | 0.912 | min: | 0.983 | min: | 0.746 |
| | max: | 2.40 | max: | 2.64 | max: | 2.20 |
| MISR SSA (558 nm) | mean: | 0.914 | mean: | 0.909 | mean: | 0.905 |
| | $\pm\sigma$: | $\pm0.0322$ | $\pm\sigma$: | $\pm0.0295$ | $\pm\sigma$: | $\pm0.0324$ |
| | min: | 0.799 | min: | 0.812 | min: | 0.799 |
| | max: | 0.987 | max: | 0.984 | max: | 0.986 |
| MISR BlS fraction | mean: | 44.4 | mean: | 50.3 | mean: | 53.7 |
| (% total AOD) | $\pm\sigma$: | $\pm21.3$ | $\pm\sigma$: | $\pm20.6$ | $\pm\sigma$: | $\pm21.0$ |
| | min: | 0.0 | min: | 0.0 | min: | 0.0 |
| | max: | 100 | max: | 96.2 | max: | 100 |
| MISR BrS fraction | mean: | 7.05 | mean: | 6.39 | mean: | 4.71 |
| (% total AOD) | $\pm\sigma$: | $\pm10.7$ | $\pm\sigma$: | $\pm12.3$ | $\pm\sigma$: | $\pm8.67$ |
| | min: | 0 | min: | 0 | min: | 0 |
| | max: | 48.2 | max: | 73.9 | max: | 6.24 |
| MISR BlS ratio | mean: | 85.6 | mean: | 88.1 | mean: | 91.0 |
| (%, BlS: BlS + BrS) | $\pm\sigma$: | $\pm21.1$ | $\pm\sigma$: | $\pm20.20$ | $\pm\sigma$: | $\pm16.3$ |
| | min: | 0 | min: | 0 | min: | 0 |
| | max: | 100 | max: | 100 | max: | 100 |
| MISR non-absorbing fraction | mean: | 34.6 | mean: | 31.8 | mean: | 29.2 |
| (% total AOD) | $\pm\sigma$: | $\pm19.0$ | $\pm\sigma$: | $\pm17.2$ | $\pm\sigma$: | $\pm19.0$ |
| | min: | 0.0 | min: | 0.0 | min: | 0.0 |
| | max: | 89.3 | max: | 80.9 | max: | 90.0 |
| MISR nonspherical fraction | mean: | 1.53 | mean: | 0.949 | mean: | 0.918 |
| (% total AOD) | $\pm\sigma$: | $\pm4.86$ | $\pm\sigma$: | $\pm3.30$ | $\pm\sigma$: | $\pm3.38$ |
| | min: | 0 | min: | 0 | min: | 0 |
| | max: | 40.0 | max: | 21.7 | max: | 9.4205 |
| Number of plumes | 223 | | 215 | | 225 | |

ability in plume heights except for a sharp drop in September ($p < 0.05$) and overall higher heights in June ($p < 0.05$ except when compared to the month of May). The greater heights in June may be partly driven by higher PBL heights (Table 2b), whereas the lower heights in September are likely driven at least in part by colder temperatures at this time of year, which are not as conducive to intense burning and vertical plume development. The relatively lower median plume FRP in September (38 W m$^{-2}$) supports this interpretation (Table 2b). Overall, the highest-altitude plumes tend to be as-

sociated with the highest cumulative plume FRP values (i.e., the sum of all fire pixels in the plume, as distinct from the median value discussed above; Fig. S1 in the Supplement), although there is considerable variability ($r^2 = 0.37$). This weak relationship is consistent with similar studies, including but not limited to Gonzalez-Alonso et al. (2019) and Val Martin et al. (2010, 2012 TS14).

The MERRA-2 data indicate that PBL heights are centered $\sim 1.5$ km above the surface on average, with the deepest layers during the summer months of June, July, and August (small differences between these 3 months are not statistically significant, whereas the lower values in both May and September are distinct, with $p < 0.05$). We estimate that 22 % of plumes were injected above the boundary layer, using a conservative criterion that the median plume height must be at least 100 m greater than the MERRA-2 PBL height to qualify as reaching to free troposphere (FT). There is a strong seasonal component to the percent of plumes injected into the FT, steadily decreasing from month to month (Fig. 2d, Table 2b). Warming temperatures increase PBL heights between May and June, which contributes to diminishing the likelihood that plumes will reach or exceed the bottom of the FT. Colder temperatures then decrease PBL heights in September, but plumes are the *least* likely to reach the FT during this month. This is likely at least partially driven by the differences in the median plume FRP and plume heights discussed above – with consistent FRP and height values across June, July, and August as well as a significant decrease in both values come September. In addition, the strength of the temperature inversion at the top of the PBL is $\sim 50$ % higher for plumes observed in September compared with earlier months, so atmospheric stability is likely an important factor constraining plume vertical development at this time of year (Table 2b). Figure 2 illustrates the seasonal variability in the plume count and FT injection for all three fire types. We find no significant differences in the height of the boundary layer or above ground level plume heights between the three types, which reinforces the idea that the ambient atmospheric structure combined with month-to-month variations in fire intensity are the dominant factors affecting wildfire plume rise in this region.

The smoke plumes injected into the free troposphere exhibit significantly larger FRP values than those confined to the boundary layer ($p \ll$ TS15 $0.05$), as might be expected. Median plume FRP values are centered at 66 and $42 \, \mathrm{W \, m^{-2}}$ for plumes above and below the top of the PBL, respectively, and cumulative FRP values are centered at values of 2502 vs. $629 \, \mathrm{W \, m^{-2}}$. Plumes in the FT were also associated with relatively weaker temperature inversions at the top of the PBL compared with those not in the FT, although both mean values still indicate positive stability overall ($3.53 \, \mathrm{K \, km^{-1}}$ vs. $5.34 \, \mathrm{K \, km^{-1}}$ on average; $p \ll 0.05$). Together, this is consistent with the prevailing theory that a combination of fire intensity and atmospheric structure are the important factors modulating the smoke plume vertical distribution. Of

the plumes injected above the PBL, we found that 48 % were associated with distinct stable layer in the FT and that plumes in the FT associated with stable layers exhibited lower heights than those not associated with stable layer layers (median heights of 1.83 km vs. 2.01 km, and max heights of 2.87 vs. 3.17 km; $p < 0.05$). This follows from the results of Kahn et al. (2007) TS16 and Val Martin et al. (2010), who showed that plumes injected above the boundary layer are largely concentrated within layers of relative stability in the FT.

## Differences among fire types

TS17 Fires identified in forest (F) tend to have cumulative plume FRP values that are essentially twice as high as other biomes ($p \le 0.014$; Table 4). The fire pixel count for these plumes (i.e., estimated fire area) likely drives the latter, as the average number of hotspots in these fires also outnumbers those in the other biomes by about $2:1$. In fact, the *median* per-pixel FRP for F fires is relatively low (only $40 \, \mathrm{W \, m^{-2}}$ on average), whereas both W and G fires exhibit somewhat higher median FRP values (55 and $44 \, \mathrm{W \, m^{-2}}$ on average, respectively), with W fires showing the largest FRP values. The median per-pixel FRP values for the F fires that we observed also never surpassed $230 \, \mathrm{W \, m^{-2}}$, whereas the W and G fires were observed to reach median FRP values of 367 and $307 \, \mathrm{W \, m^{-2}}$, respectively. This is consistent with previous studies of forest fires which showed that a key fuel component is at or below the surface, as roots systems allow fires to burn deeper into soil layers compared with biomes dominated by finer fuels, such as grass and savanna (e.g., Gonzalez-Alonzo et al., 2019). The higher moisture content and lower oxygen availability in these subsurface layers are more conducive to smoldering than the flaming fire phase, which leads to higher smoke production (AOD) but lower radiant emissivity (Bertschi et al., 2003; Yokelson et al., 1997; Gonzalez-Alonso et al., 2019; van der Werf et al., 2010; Santoso et al., 2019). Therefore, the comparatively lower average per-pixel FRP in F fires is consistent with a higher fraction of smoldering, and the higher FRP values in W and G fires are consistent with flaming conditions dominating. The idea that smoldering is favored in increasingly forested areas is also supported by significantly higher MISR AOD in F ($\sim 1.54$) and W ($\sim 1.45$) plumes compared with G ($\sim 1.15$) plumes ($p < 0.001$). (More information on particle properties is presented in subsequent sections.) However, the differences in the per-pixel FRP between F and G fires were not found to be statistically significant ($p = 0.35$), whereas the differences between W fires and F or G fires were significant ($p < = 0.025$). It is possible that W fires have the highest median FRP because they have a lower fraction of smoldering than F fires but burn longer than G fires and, therefore, more completely fill the satellite fire pixels. This hypothesis might be tested with higher-spatial-resolution FRP observa-

Atmos. Chem. Phys., 22, 1–24, 2022

https://doi.org/10.5194/acp-22-1-2022

tions than are available for the current study (e.g., from aircraft measurements).

## 3.2 Overview of smoke particle property observations

The observed smoke plumes exhibit a wide range of MISR-retrieved light-absorption properties, with the median plume mid-visible SSA ranging all the way from $\sim 0.8$ to 1.0 and the fractional AOD of BlS and BrS ranging from 0 to 1 and from 0 to $\sim 0.75$, respectively (Table 4). Differences in plume REPA strongly drive the interpreted BlS content ($r^2 = 0.7$), whereas fractions of BrS are not correlated with retrieved light absorption ($r^2 = 0.1$: Fig. S2 in the Supplement). The MISR-retrieved particle size is somewhat less variable than light absorption, with all plumes but one exhibiting a median ANG of at least 1.0 (fine-mode particles) and the highest plume exhibiting an ANG value of just below 2.7. Most plumes have retrieved ANG values of 1.5–2.0.

To help interpret the ANG, we analyze each of the four particle size bins defined in the RA climatology (Tables 1 and S1 in the Supplement) in terms of their fractional contribution to the total AOD, where very small particles have an effective radius ($r_e$) of $\sim 0.06\,\mu m$, small particles have an $r_e$ of $\sim 0.12\,\mu m$, medium particles have an $r_e$ of $\sim 0.26$, and large particles have an $r_e > 1.21\,\mu m$ (large particles include the nonspherical particle type as well as spherical types). To appropriately interpret the MISR-retrieved size constraint, we refer to the retrieved effective particle size (REPS), which indicates qualitative changes in the effective size of the retrieved mixture of particle types. The MISR REPS aggregates the contributions of the different size components, and the retrieved ANG, which might be more representative of the actual particle size *differences* (i.e., relative, not absolute), avoids strictly identifying one of these specific sizes. As an example, an increase in REPS corresponds to a higher AOD fraction of larger components retrieved within the plume. The actual particle size distributions are constrained within bins by the retrievals; thus, discussing size in terms of these bins is helpful. The algorithm climatology contains these sizes to capture MISR sensitivity to particle size under good but not necessarily ideal retrieval conditions, based on theoretical analysis (Kahn et al., 2001) and subsequent field validation studies. As expected, analysis of the size components in this study indicates that small and medium particles dominate the aerosol size distributions in most smoke plumes regardless of fire type, on average constituting $\sim 46\%$ and $\sim 25\%$ of the median plume AOD, respectively. Very small particles make up $\sim 11\%$ of retrieved plume AOD on average, whereas large particles make up $\sim 3\%$ (Fig. 3a). In most plumes, the fraction of nonspherical particles is low, constituting less than 5% of the total retrieved AOD in 93% of plumes (Table 4).

## 3.3 Impact of land cover type on smoke particle properties

Fires detected in forests (F) have the highest overall AOD values (1.54), whereas woody (W) and grassy (G) fires exhibit an average AOD of 1.45 and 1.15, respectively (Table 4). F plumes also have the highest fraction of nonspherical particles, with a few plumes containing as much as 40% nonspherical particles, although on average these soil or dust analogs make up only $\sim 1.5\%$ of the total AOD, and differences in this fraction between plume types were not found to be statistically significant. However, the higher relative contribution of nonspherical particles (which are larger in size than the spherical MISR components; see Table 1) in F plumes would partially account for their overall lower ANG (1.65) compared with W (1.70) and G (1.75) plumes. These differences in REPS, although small, are significant ($p \leq 0.038$) in all three inter-biome comparisons. Figure 3 suggests that the main driver of differences in particle size between plume types is the partitioning between small and medium particles, as the fractional contribution from very small particles is nearly identical across fire type, and differences in the fraction of large particles are only on the order of a few percent of total AOD. In F plumes, the contribution from medium particles is highest (29%), although small particles still dominate (41%). In W plumes, these fractions favor small particles significantly more (47%, $p \ll 0.05$). The greatest difference can be seen in G plumes, with nearly 51% of particles in the small category on average and only a 20% contribution from medium particles. The extent to which these differences occur during emission vs. downwind aging are explored in Sect. 3.4.

Particles are less light absorbing in F fires (SSA $\sim 0.914$) compared with G fires (0.904) ($p = 0.002$), whereas W fires have SSA values between these two (0.909). All three fire types display similar ranges of possible values (Table 4). Analysis of the individual contributions from moderately to strongly light-absorbing components (BlS and BrS), weakly absorbing components (nonspherical), and non-absorbing components suggests that differences in the plume-averaged REPA manifest in a combination of (1) the *difference* in the fractional AOD of non-absorbing vs. moderately to strongly absorbing aerosols; (2) to a lesser degree, the *partitioning* of BlS vs. BrS; and (3) the *relative rate of change* in particle type with plume age. To a certain degree, REPS also drives these factors, as larger particles with the same composition will have lower SSA. The first two factors can be seen in Table 4, with the sum of BlS and BrS being higher in G fires compared with F fires as well as especially high fractions of BlS in G fires and BrS in F fires ($p < 0.001$ in both cases). W fires are not considered statistically different from F or G fires in this regard, and they exhibit values between these two. These findings are consistent with other studies, as smoldering fires (which are more frequent in forest compared with grassland and savanna fires) tend to pro-

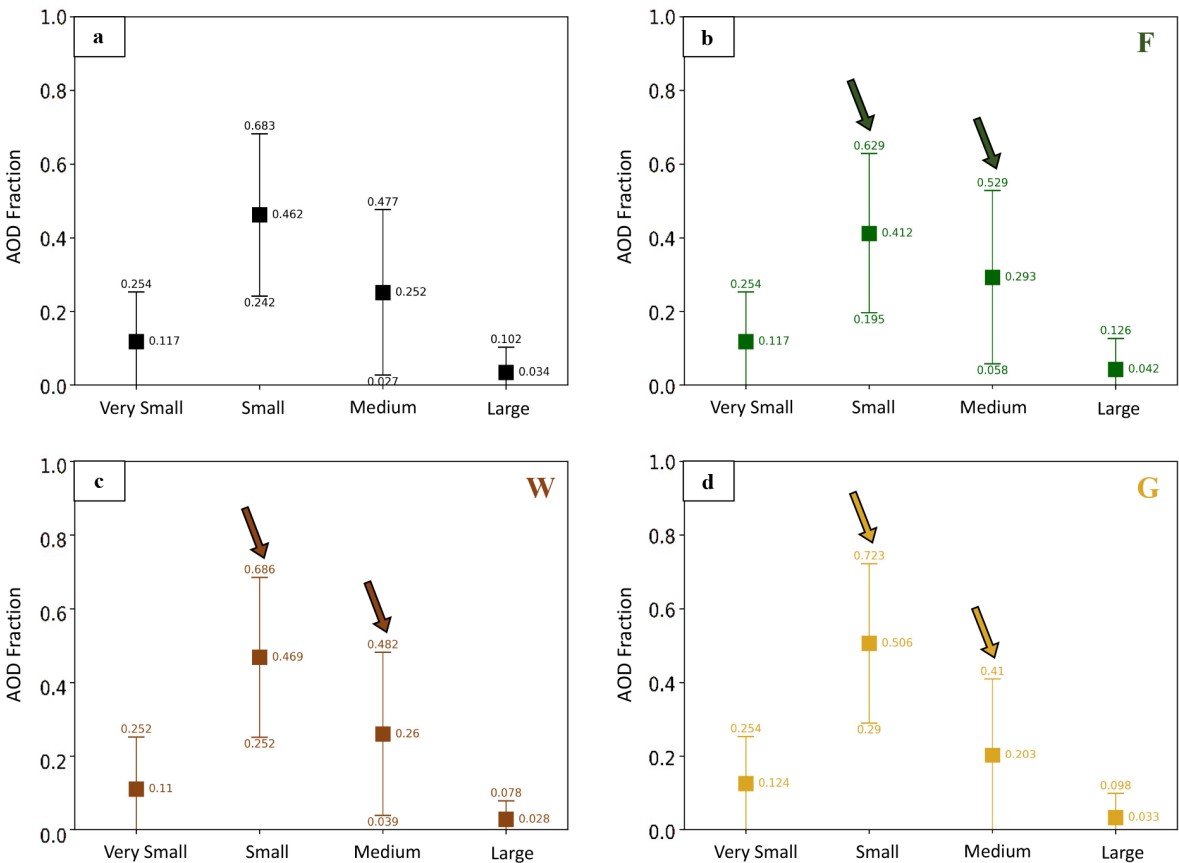

**Figure 3.** Particle size distributions in terms of each RA component's fractional contribution to the total AOD. All plume types are shown in panel **(a)**, whereas panels **(b)**–**(d)** display color-coded results for individual plume types, denoted with the appropriate abbreviation in the upper right-hand corner. Points represent the mean values, and whiskers show the standard deviations. Arrows highlight the differences in the partitioning between the small and medium particle fractions for the different plume types.

duce higher fractions of BrC particles compared with flaming fires, but fires in savannas and grasslands emit larger fractions of BC (Chakrabarty et al., 2010, 2016; Petrenko et al., 2012). The third factor driving inter-biome REPA differences is explored in Sect. 3.4, in our discussion of particle aging.

### 3.4   Downwind particle evolution and differences between fire types

To understand how particle properties change with smoke age both within and between plumes, we use MINX wind speeds and distance from the fire source to divide each plume into discrete age bins at approximately 30 min intervals, where possible. In 6 % of cases, poor retrieval quality and/or gaps in the retrieved plume area prevented us from calculating age in a plume.

Overall, REPS increases (lower ANG) whereas REPA decreases (higher SSA) with age. This is generally consistent with the literature on the typical aerosol aging processes in BB plumes, as particles oxidize and hydrate, and gaseous precursors such as volatile organics condense onto their sur-

face to increase particle size; these processes often lead to reduced light absorption, especially in the case of BlS. As coatings generally increase hygroscopicity, this can contribute to increased particle hydration, size and SSA, especially as the plume cools, which increases the effective relative humidity CE13. However, we find that plumes in the G category exhibit *decreased* REPS with age, unlike the trends seen in F and W plumes (Fig. 4a–c). This is accompanied by a dramatic decrease in the downwind plume AOD not seen in F or W plumes (Fig. 4d). In Fig. 5, trends in REPS are further illustrated via the downwind evolution in the AOD fractions of the four particle-size components. In particular, the partitioning between AOD fractions of small and medium particles is highlighted. We observe a transition from smoke dominated by small particles to smoke dominated by medium particles at approximately the 3 h mark in both F and W plumes, whereas G plumes never experience this transition and, in fact, tend to show the opposite CE14. In the context of the total plume AOD decreasing, this may indicate that dilution by background air is more important in G plumes than in F and W plumes. In contrast, F and W plumes appear

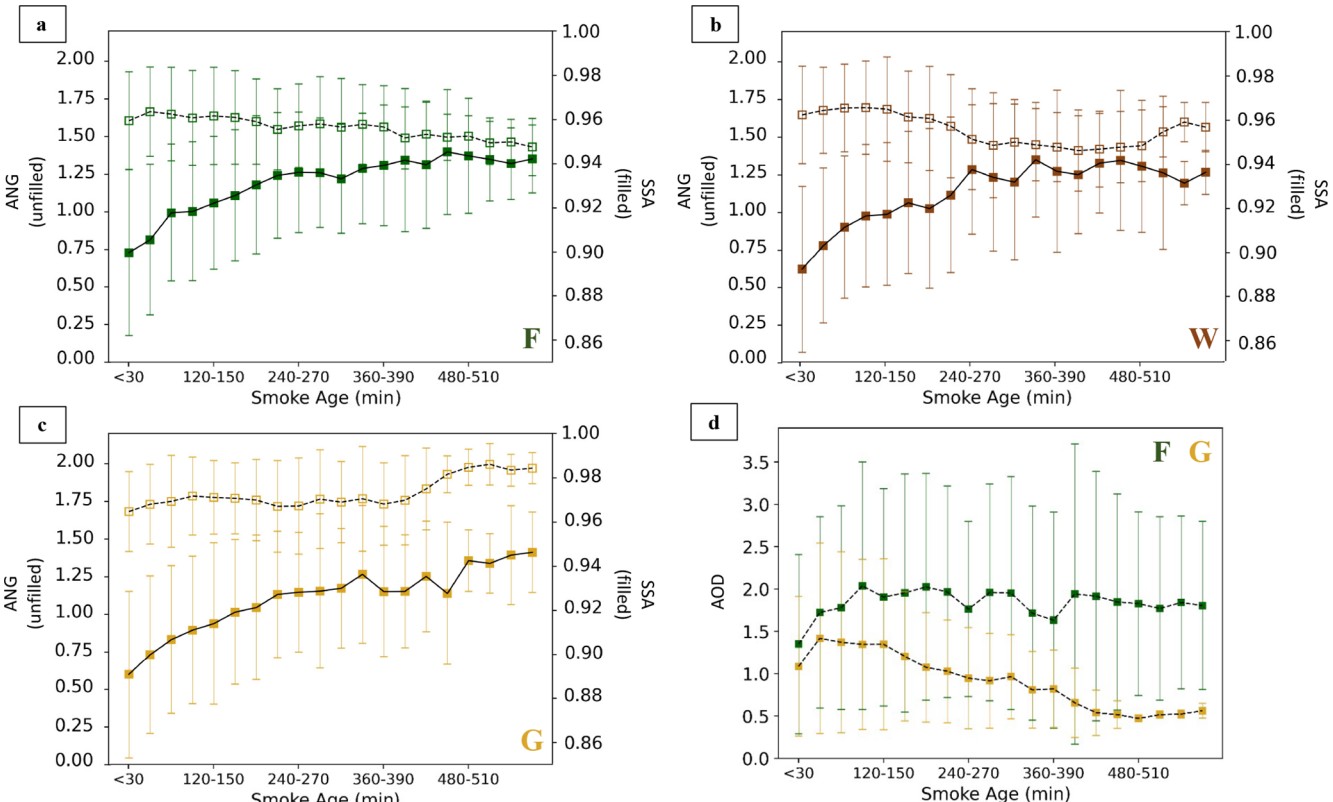

**Figure 4.** MISR mid-visible Ångström exponent (unfilled markers, dotted lines, left vertical axes) and single-scattering albedos (filled markers, solid lines, right vertical axes) by smoke age for **(a)** forest plumes, **(b)** woody plumes, and **(c)** grassy plumes. (The fuel type is indicated in the lower right-hand corner of these plots.) In panel **(d)**, the MISR mid-visible AOD is plotted by age for forest and grassy plumes. The points represent the mean values, and the whiskers are standard deviations.

to experience downwind particle growth due to a dominance of condensation/hydration. The decreased AOD and particle size may also indicate that G plumes experience a stronger shift in gas–particle partitioning downwind, as the mixing of cleaner background air into the plume shifts the equilibrium for semi-volatile compounds from the particle phase to the gas phase, resulting in stronger rates of evaporation at lower concentrations (Garofalo et al., 2019).

Although plume REPA decreases downwind in all three plume types, the timescales over which the particle-type components transition from absorbing-dominated to non-absorbing-dominated components are significantly different between G plumes and the other two plume types. Non-absorbing particles begin to dominate over the BlS components at approximately the 3 h mark in F and W plumes, whereas the transition takes between $\sim 4$ and $\sim 7$ h in G plumes. The longer retrieved lifetime for BlS in G plumes may indicate reduced levels of oxidation, which is consistent with the fact that flaming fires emit less VOCs that are important in modulating gas-phase oxidation chemistry (Liu et al., 2017; Koppmann et al., 2005; Donahue et al., 2014). This would also help account for smaller particle sizes in G

plumes, as VOC condensation is an important mechanism for particle growth.

## 3.5 Impact of drought on particle properties

The majority of plumes (65 %) burned during drought or abnormally dry conditions, with F and W plumes tending to respond to drought more than G plumes overall (Fig. 7). In F plumes, AOD tends to increase with drought index once drought becomes severe, whereas moderate drought or abnormally dry conditions do not produce a significant response in AOD (compared to normal conditions). In W plumes, no significant trend emerges between AOD and drought index except when comparing normal conditions to the most extreme drought levels, at which point plume AOD is significantly higher. In contrast, G plumes experience no change in AOD with increasing drought (Fig. 8). These differences are consistent with the findings of other studies on fuel consumption across different biomes: areas with a low tree cover density (e.g., grasslands) experience lower tree mortality rates during drought stress compared with areas with a high tree cover density (van der Werf et al., 2010). Therefore, we expect forested areas to experience

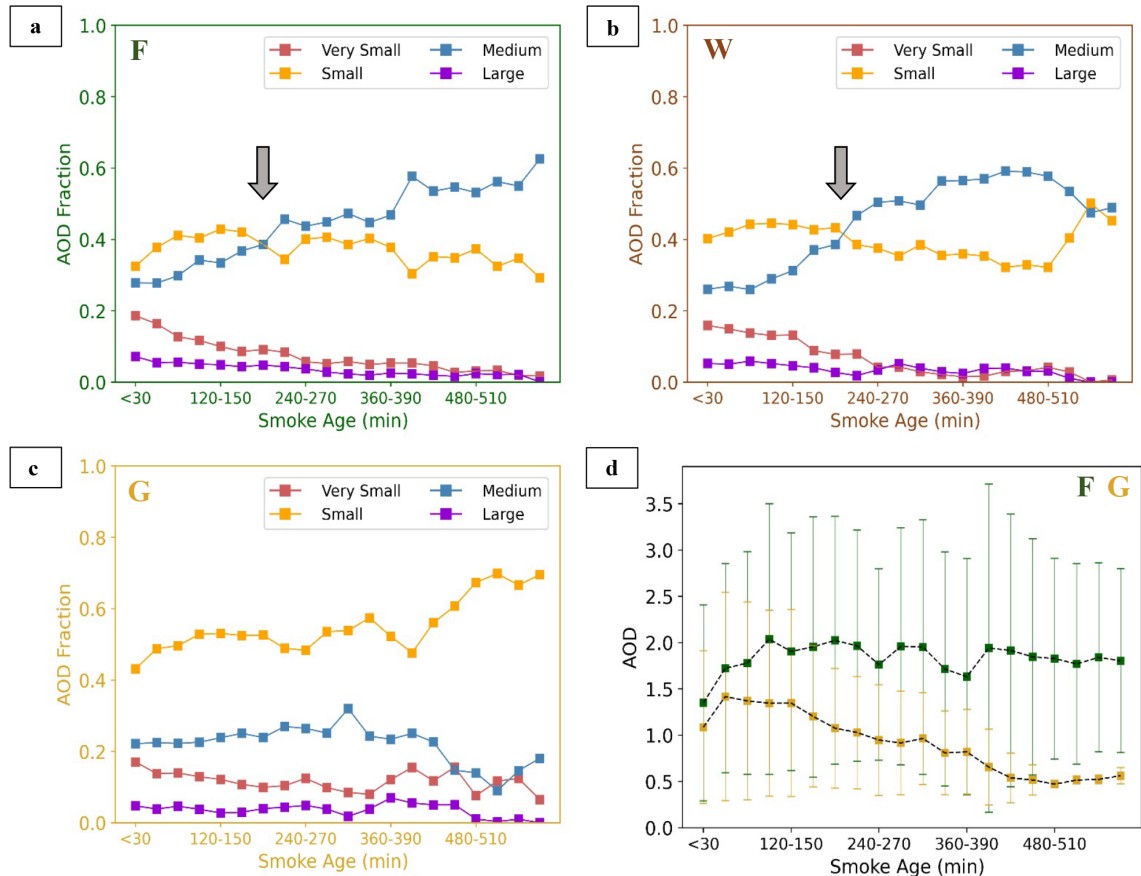

**Figure 5.** MISR particle-size component AOD fractions (in terms of the contribution to the total AOD, from 0 to 1) by smoke age for **(a)** forest plumes, **(b)** woody plumes, and **(c)** grassy plumes. In panel **(d)**, the MISR mid-visible AOD is plotted by age for forest and grassy plumes, for reference. The points represent the mean values, and the whiskers are standard deviations. Arrows help highlight the points of important particle-size transitions.

larger swings in the amount of fuel available for burning based on drought, compared with low-biomass-density areas. The considerable increases in fuel availability translate to larger, more intense fires in the W and especially F categories. In fact, the cumulative FRP over the burning area increases with drought in F and W fires, corresponding to a larger number of fire detections per plume.

We find that increasing drought index in W plumes is generally correlated with decreasing particle light absorption (higher SSA) and BlS fractions, trends that are not present in F and G plumes (Fig. 9). In addition, both W and G plumes burning during normal conditions (the "None" category in Figs. 8–9) exhibit higher fractions of BlS than F plumes burning in normal conditions ($p < 0.05$). These differences may indicate that shifts in fuel type and burning regimes are more pronounced in W plumes. A possible explanation is that fires identified as W plumes experience some of the largest swings in the types of fuels available for burning, as they contain a more even mixture of grassy vs. woody vegetation based on our definitions. As drought persists, the relative increase in the amount of woody fuels available to burn in W

fires outpaces that in F fires, which consume a larger fraction of woody fuels to begin with, as per our fire-type definitions. Similarly, the absence of coarse woody fuels in grassy biomes would prevent any significant shift in fuel types. It would then stand to reason that W plumes experience the most significant shift in the fraction of smoldering to flaming regimes as drought becomes more severe, which is supported by the decreasing fraction of BlS in Fig. 9d (and, therefore, an increase in the fraction of BrS emissions that tend to be associated with smoldering fires). However, no statistically significant trends in median nor mean plume FRP were detected in any of the three plume types. Still, the inherent limitations of using FRP to determine fire strength and burning regime means the possibility cannot be ruled out.

## 4   Conclusions

This work, focused on Canada and Alaska, represents the first regional study in an ongoing effort to characterize wildfire particles across the globe. Thanks to extensive valida-

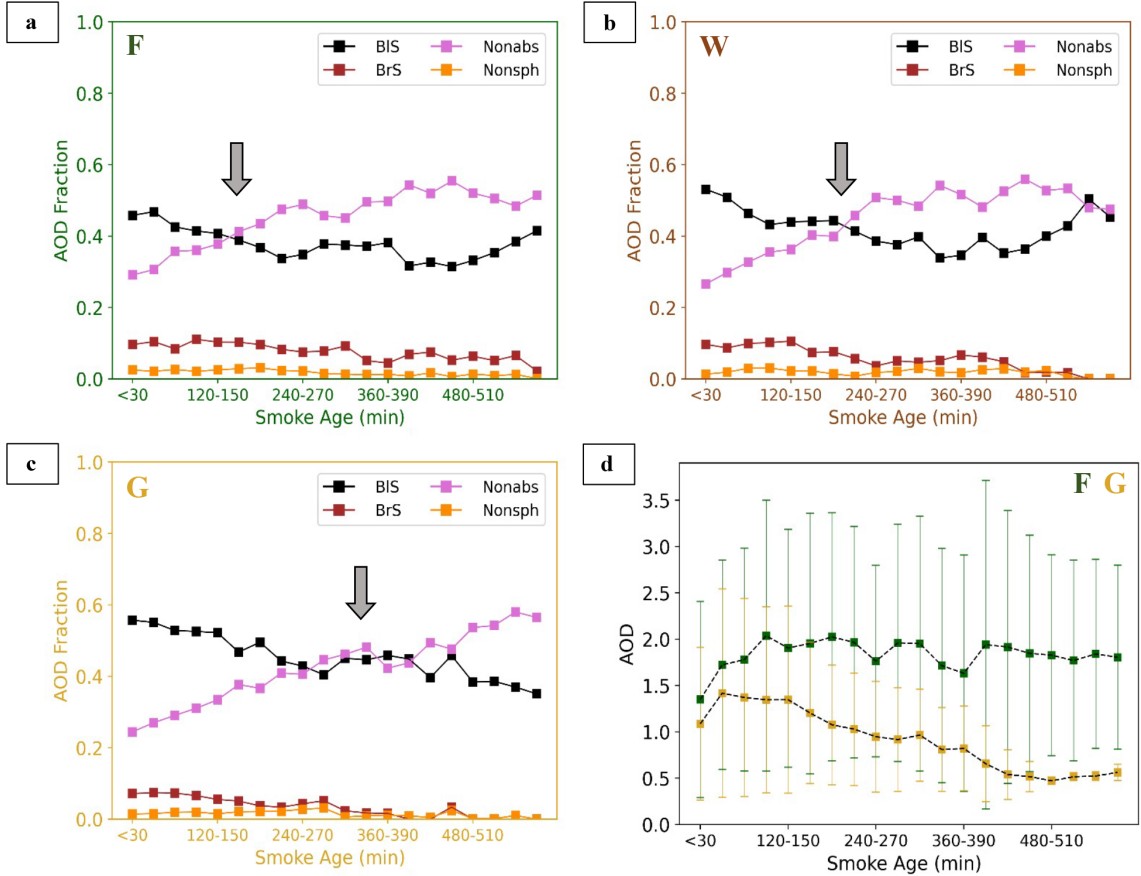

**Figure 6.** MISR particle-type component AOD fractions (in terms of the contribution to the total AOD, from 0 to 1) by smoke age for **(a)** forest plumes, **(b)** woody plumes, and **(c)** grassy plumes. In panel **(d)**, the MISR mid-visible AOD is plotted by age for forest and grassy plumes, for reference. The points represent the mean values, and the whiskers are standard deviations. Arrows help highlight the points of important particle-size transitions. TS18

tion experiments in previous work, we apply our knowledge about the relationships between particle chemistry/microphysics and the TOA optical signatures obtained by MISR to create a regional inventory of particle type (black smoke vs. brown smoke vs. soil/dust vs. non-light-absorbing particles); inferred trends in particle evolution (e.g., oxidation, size-selective dilution, and hydration/condensation); and the modulating forces behind these, such as meteorology, land cover type, and fire intensity. The MISR plume heights and particle properties exhibit patterns that match the existing literature on the role of burning conditions and vegetation in wildfire smoke properties well. Furthermore, the *combination* of MISR observations with other satellite and modeling datasets allows us to infer the dominant factors driving particle properties and their evolution under different conditions, along with the associated timescales. This represents new territory in BB aerosol studies.

Specifically, we find distinct patterns in plume properties when the data are partitioned into three categories based on the relative fractions of forest (F), woody savanna (W), and grasslands/savannas/shrublands (G). The largest differences are typically found between F and G fires, as these represent the extremes in fuel type for fires in the study region. The most statistically significant differences are observed in (1) MISR AOD, with thicker plumes in the F category, and (2) particle size and light absorption, with F plumes exhibiting both larger and less-absorbing particles than G plumes. These differences are likely driven at least partially by the relative fractions of flaming and smoldering fire in each category, as smoldering is more dominant in F fires.

There also appear to be distinct differences in how smoke particles age downwind with plume type as well as the timescales over which these changes occur. In G plumes, particles are not observed to experience increases in particle size, it takes comparatively longer for the BlS AOD fraction to diminish, and total AOD drops significantly downwind. In F and W plumes, the near-source dominance of small particles transitions toward medium particles, non-absorbing particles begin to dominate over BlS particles much sooner, and total AOD is relatively consistent downwind. Based on these trends, we infer that G plumes and F/W plumes experience

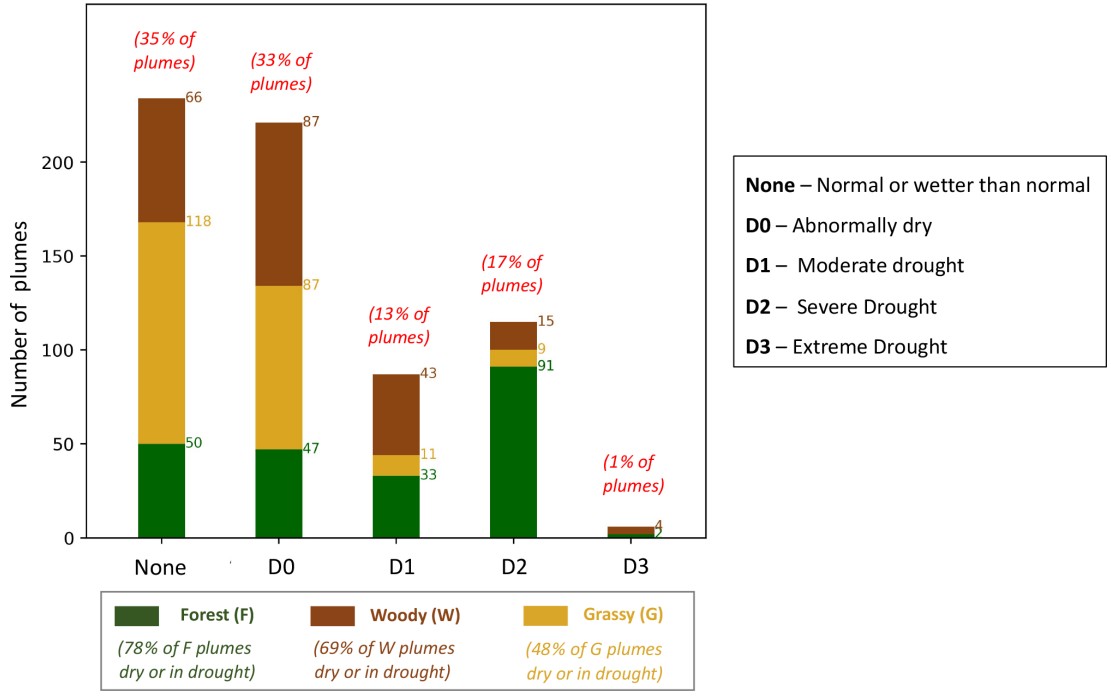

**Figure 7.** The number of plumes observed in each drought category (see key), colored by plume type.

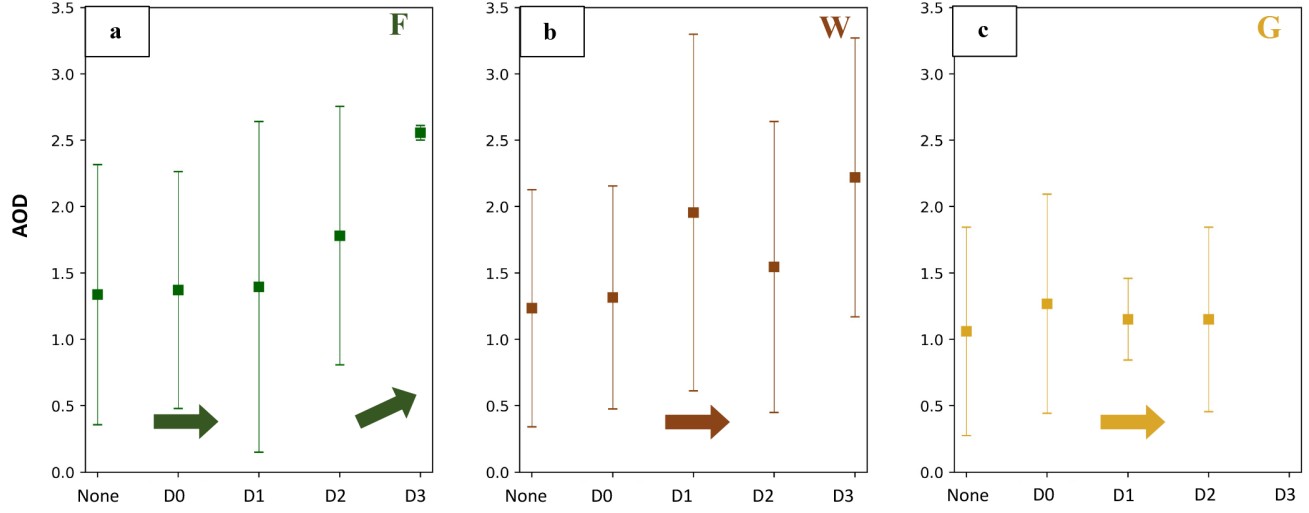

**Figure 8.** Median plume AOD according to plume type and drought level. Panels are color-coded by plume type and identified with the appropriate abbreviation in the top right-hand corner. Points represent the mean values, and whiskers are the standard deviations. Arrows highlight the general trends with increasing drought: up arrows signify that the trend is statistically significant, and flat arrows indicate that they are not. Note that AOD begins to increase with increasing drought level from D2 onward for forest plumes; for woody plumes, the differences in AOD are only significant when comparing the "None" category to the D3 category, with no clear trend in intermediate categories; for grassy plumes, there are no statistically significant relationships between AOD and drought severity.

varying types and degrees of atmospheric aging. Namely, we infer the following:

1. G plumes experience less oxidation and condensation compared with F and W plumes, evidenced by the higher overall absorbing aerosol fraction retrieved by

MISR and the fact that flaming fires produce fewer VOC emissions;

2. dilution may play a larger role in particle size for G plumes, supported by decreasing particle sizes downwind and reduced AOD downwind.

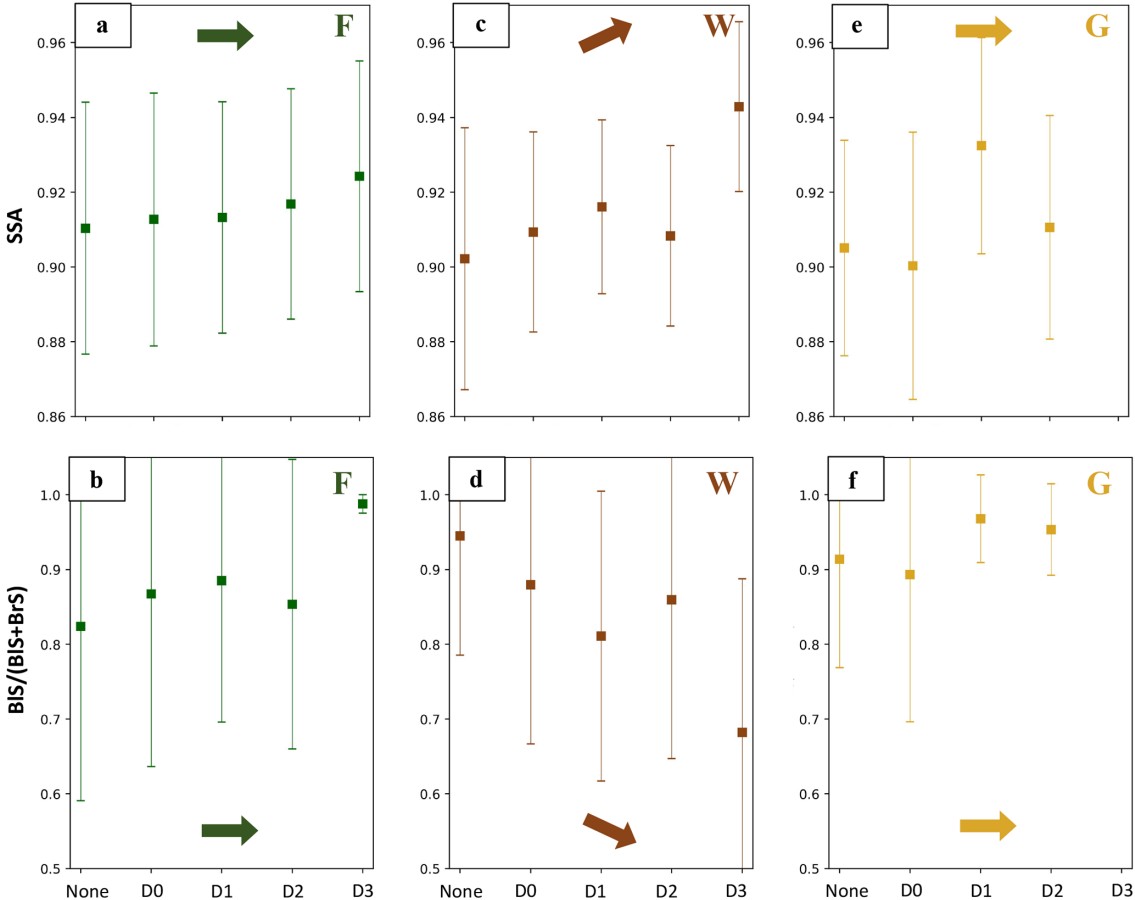

**Figure 9.** MISR mid-visible single-scattering albedo **(a, c, e)** and the fraction of AOD from BlS vs. the sum of both BlS and BrS **(b, d, f)** for different drought conditions. Decreasing values indicate lower amounts of BrS and higher amounts of BlS. Panels are color-coded by plume type and identified with the appropriate abbreviation in the top right-hand corner. Points represent the mean, and whiskers are the standard deviations. Arrows help highlight the general trends with increasing drought index: up or down arrows signify that the trends are statistically significant, and flat arrows indicate that they are not. Note that the D2 category presents an outlier for the otherwise clear trends in SSA and BlS / (BlS + BrS) for woody plumes; for F and G plumes, there is no statistically significant response to increasing drought severity for either variable.

These conclusions, made possible by the extensive coverage and extensive sample size provided by satellite remote sensing, represent new territory, as there have been no other studies (to date) of such scope relating particle properties to specific aging mechanisms and the timescales over which they occur. CE15

We also find that drought plays a role in AOD for F plumes (and, to a lesser extent, W plumes), with higher drought stress leading to increased plume AOD. However, only W fires appear to respond significantly to drought in terms of the smoke particle properties, with particles becoming brighter with increasing drought stress. These results suggest that F plumes are more susceptible to drought-induced changes in fuel *amount*, whereas W plumes respond more to changes in the fuel *type*. Our results also indicate that G fires are resistant to changes from drought, which is consistent with current expectation.

Future work will involve applying the MISR RA to plumes across a wide variety of other biomes and climate conditions. Based on current knowledge of the differences in fire properties between regions, we expect to find significant differences in particle properties, plume heights, and other retrieved or modeled quantities, with implications for the underlying mechanisms and the timescales over which they operate. Such observational constraints on BB particle-type distributions and aging regimes could greatly benefit regional and global climate and air quality modeling efforts.

**Code and data availability.** The MISR Research Aerosol (RA) algorithm is a proprietary product. MINX is available for public use TS19. The RA and MINX results for individual plumes can be found at the NASA Langley Atmospheric Data Center (ASDC) Distributed Active Archive Center (DAAC): [final address is TBD] TS20.

https://doi.org/10.5194/acp-22-1-2022 Atmos. Chem. Phys., 22, 1–24, 2022

Please note the remarks at the end of the manuscript.

**Supplement.** The supplement related to this article is available online at: https://doi.org/10.5194/acp-22-1-2022-supplement.[TS21]

**Author contributions.** The project was first conceptualized by RAK, and the development and design of the methodology were a collaboration between RAK and KTJN. The RA algorithm used in this project was developed by RAK and JAL, and it is maintained by JAL. KTJN developed the tools used to process, analyze, and visualize all data. Formal analysis of the results was conducted by KTJN and RAK, who together wrote the original draft. ZL reviewed and provided comments on the manuscript. All authors read and agreed to the final version of the paper.

**Competing interests.** At least one of the (co-)authors is a member of the editorial board of *Atmospheric Chemistry and Physics*.

**Disclaimer.** Publisher's note: Copernicus Publications remains neutral with regard to jurisdictional claims in published maps and institutional affiliations.

**Acknowledgements.** The authors thank Steven F. Noyes for his assistance with the development of the software tools used for data processing and analysis. We also acknowledge the use of imagery from the NASA Worldview application (https://worldview. earthdata.nasa.gov), which is part of the NASA Earth Observing System Data and Information System (EOSDIS).

**Financial support.** The work of Katherine T. Junghenn Noyes is currently supported by the NASA Postdoctoral Program (NPP), managed by the Universities Space Research Administration (USRA). Part of this work was conducted during Katherine T. Junghenn Noyes' PhD studies, which was supported by a grant from the Maryland Space Grant Consortium under Richard C. Henry and Matt Collinge, as well as NASA's Atmospheric Composition Modeling and Analysis Program, under Richard Eckman through a grant to Ralph A. Kahn, and a NASA grant (NNX16AN61G) that supports Zhanqing Li. The work of Ralph A. Kahn and James A. Limbacher is supported in part by the NASA Climate and Radiation Research and Analysis Program under Hal Maring, the NASA Atmospheric Composition Modeling and Analysis Program under Richard Eckman, and the EOS Terra and MISR projects.

**Review statement.** This paper was edited by Yves Balkanski and reviewed by two anonymous referees.

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

## Remarks from the language copy-editor

CE1  Is "NASA Postdoctoral Program" a part (e.g., department) of this institution? If not, it should not be mentioned in this section. Please advise.

CE2  Please confirm the change throughout the paper.

CE3  Please note that, according to our standards, we always define abbreviations at the first instance of use (in the abstract as well as in the rest of the text).

CE4  Please note that, according to our house standards, units not preceded by a number (outside of parentheses) must be written out in full. Please check all instances throughout. Also, are the abbreviations outlined in footnote *a* actually required? They do not seem to be used in the table or in the paper. Please check.

CE5  Please note that the ACS recommends the use of "Å" to represent the Ångström exponent. Should this be changed throughout?

CE6  Is "True Color" referring to specific imagery or is it a descriptive term? Please advise.

CE7  Please note that "Yuk." seems to be missing from this footnote. Should it be added?

CE8  Please note that Copernicus Publications requires the sole use of 24-hour time.

CE9  Please confirm the change.

CE10  Please confirm the change. Please note that "mbar" is our house standard for millibars.

CE11  Do you mean "superimposed"?

CE12  Do you mean "large statistical sample"?

CE13  Please confirm the change.

CE14  Please check that the meaning of your sentence is intact.

CE15  Please check that the meaning of your sentence is intact.

## Remarks from the typesetter

TS1  The composition of all figures has been adjusted to our standards. All figures were taken from the authors manuscript *.pdf file for quality reasons. Please confirm. Please also note the hyphen removal in the key figure.

TS2  Please confirm or provide a different short running title.

TS3  Please provide year.

TS4  Please provide year.

TS5  Please provide date of last access.

TS6  This reference is not in the reference list. Please add it.

TS7  This reference is not in the reference list. Please add it.

TS8  Please note that units have been changed to exponential format. Please check all instances.

TS9  Please provide year.

TS10  Please provide year.

TS11  Please provide date of last access.

TS12  Please add this information to the data availability section and provide a corresponding reference list entry (including creators, title, repository/publisher, and date of last access)

TS13  Please note that "a.g.l." is our standard for "above ground level".

TS14  This reference is not in the reference list. Please add it.

TS15  Please confirm ≪.

TS16  This reference is not in the reference list. Please add it.

TS17  Please note that, in keeping with our house standards, this section remains unnumbered, as there is no Sect. 3.1.2.

TS18  Please mention Fig. 6 in the text.

TS19  Please indicate how MINX can be obtained/accessed.

TS20  Please provide a direct link to the dataset and, if possible, a DOI instead of a URL. In any case, please provide a reference list entry including creators, title, and date of last access or indicate which specific reference entries are meant.

TS21  Please send a new supplement as a *.pdf without the title, authors, correspondence author, etc. as we will generate a supplement title page during publication (with a citation including the DOI), which will contain this information.

TS22  Please ensure that any datasets and software codes used in this work are properly cited in the text and included in this reference list. Thereby, please keep our reference style in mind, including creators, titles, publisher/repository, persistent identifier, and publication year. Regarding the publisher/repository, please add "[dataset]" or "[code]" to the entry (e.g., Zenodo [code]).

TS23  Please provide date of last access.

TS24    Please provide all author names and make sure that all authors are listed in the correct order: last name, initial(s).

TS25    Please provide all author names and make sure that all authors are listed in the correct order: last name, initial(s).

TS26    Please provide publisher.

TS27    Please provide page range or article number.

TS28    Please provide all author names and make sure that all authors are listed in the correct order: last name, initial(s).

TS29    Please provide page range or article number.

TS30    Please provide page range or article number.

TS31    Please provide all author names and make sure that all authors are listed in the correct order: last name, initial(s).

TS32    Please provide page range or article number.

TS33    Please provide page range or article number.

TS34    Please provide all author names and make sure that all authors are listed in the correct order: last name, initial(s).

TS35    This reference is not cited in the text. Please check.

TS36    Please provide date of last access.

TS37    Please provide volume number.

TS38    Please provide page range or article number.

TS39    Please provide initials.

TS40    Please provide page range or article number.

TS41    Please provide page range or article number.

TS42    Please provide page range or article number.

TS43    Please provide page range or article number.

TS44    Please provide all author names and make sure that all authors are listed in the correct order: last name, initial(s).