# Peer review of "Canadian and Alaskan wildfire smoke particle properties, their evolution, and controlling factors, from satellite observations"

_Atmospheric Chemistry and Physics, 2021_

## Author Comment (AC1)

We thank the reviewers for helpful comments on our paper. We have addressed each comment below, and made changes to the paper as appropriate.

**RC1 Comments and Author Responses (AC)**

1. Line 201 - Isn't deposition a function of particle size? I would expect AOD and REPS to decrease with deposition as the larger particles settle out first.

   We distinguish here two types of potential deposition – one in which larger particles preferentially settle out of the column to the point where we can detect a change in REPS in the smoke plume, and one in which there is no size preference (or at least none that we can detect at the resolution of the MISR instrument). If there is sufficient turbulent mixing, for example due to wind shear, settling can be size-independent. We have also observed this in volcanic plumes (e.g., *Flower & Kahn*, JGR 2020).

2. Table 1 - is the humped SSA dependence in Table 1 expected? For example, Figure 1 in Samset et al. (2018) (https://link.springer.com/article/10.1007/s40641-018-0091-4) suggests either increasing or decreasing SSA as a function of wavelength, with both BC and BrC increasing with increasing wavelength and dust decreasing with increasing wavelength.

   The imaginary refractive indices shown in Figure 1 in Samset et al. were used as guidance to determine our own refractive indices across the different MISR wavelengths and the four size bins. Note that the assumed particle size distribution as well as the refractive indices determine the spectral SSA. Our exact values are somewhat different as we consider a variety of particle sizes (Tables 1 and S1 in the Supplement). These were then input to Mie theory calculations, which determined our final SSA values.

3. Figures 8&9 - the arrows to guide the eye - how are the arrow directions defined? is there a quantitative determination? For example, qualitatively looking at the plots I would consider drawing increasing arrows on figures 9a, 9b and 9f.

   The up or down arrows indicate that there were statistically significant trends in the data, whereas flat arrows indicate there were no statistically significant differences. We have added this clarification to the captions in Figures 8 and 9. Also, we have double checked our numbers and decided to go with a more conservative interpretation of the trends. We therefore redrew the arrows on Figure 8 and have made the appropriate changes in the text.

4. Line 44 - cite Petzold et al. (2013) www.atmos-chem-phys.net/13/8365/2013/. I like the BlS and BrS designation!

   Thanks. We now cite Petzold et al., 2013 on line 44

5. Line 87 - perhaps cite Kleinman https://doi.org/10.5194/acp-20-13319-2020 here

We now cite Kleinman et al., 2020 on line 88

6. Line 105 - 'Models are uncertain...' Models don't have thoughts...could this perhaps be rephrased?

We have rephrased this as "Models also feature substantial uncertainty…"

7. Line 118 - 'Research Aerosol (RA) retrieval algorithm' I personally would have used $RA^2$: 'Research Aerosol retrieval algorithm $(RA^2)$'. Because later, e.g., line 127 if reading aloud - 'the RA successfully mapped' reads as 'the Research Aerosol successfully mapped' would sound better as 'the $RA^2$ successfully mapped' i.e., 'the Research Aerosol retrieval algorithm successfully mapped', but no worries if RA is already accepted nomenclature

I like this idea, but 'RA' has been the nomenclature for many years now across dozens of papers. This came about because the MISR operational algorithm is called the Standard Algorithm and is "SA" is many papers.

8. Table 2 caption - AGL is only mentioned in the footnotes, so move AGL definition to the footnotes too?

We have done this.

9. Line 289 MERRA is used several times before being defined (e.g., on 137, 282, 287).

We now define MERRA-2 the first time it appears (line 137).

10. Figure 1 - Perhaps indicated the territories since they are referenced in the first paragraph of the results section?

To make the regions clear, we would also have to draw the geographical boundaries, and we think this might detract from the plotted distribution of fire locations and associated surface types. This might be a more important consideration in the current paper than the boundaries, especially as the locations of British Columbia, Alaska, and the Northwest Territories are generally known, and are mentioned only to say we tend to have high concentrations of fires there.

11. Line 348-349 - Table 3 defines the land cover types so don't need to repeat definitions here.

Agreed. This has been changed.

12. Line 363 - spell out woody to be consistent with spelling out other biomes in previous sentence.

Done.

13. Figure 2 and related discussion - Plumes in FT are the same as plumes above PBL described in Table 2 so perhaps it makes sense to change Table 2 to call them plumes in FT instead of 'above PBL'. The footnote to Table 2 could stay the same.

We phrased this using "Above PBL" in Table 2 because the table also shows PBL-Top stability and median PBL height, and being consistent within a single table seems best. We have added clarification in the text so that the reader knows "plumes in the FT" are the same as "plumes above the PBL" (Section 2.4, first paragraph)

14. Figure 2 - Some numbers in Table 2 don't quite match up with Figure 2 - for example Table 2 suggests 33.8% plumes in FT in 2016, while sum for those plumes is 32% in Figure 2. Difference due to rounding?

Yes, this is due to differences in rounding, for the sake of simpler visualization in the figure. We have clarified this in the caption for Figure 2.

15. Line 410 - the only typo I found! there is an extra 'a' at the end of this line

Thank you. This has been fixed.

16. Line 471-473 - This sentence could be clarified - possibly also split into two sentences.

The sentence previously read as: "The MISR REPS aggregates the contributions of these size components, and so the retrieved ANG, which might be more representative of the actual particle size differences (i.e., relative, not absolute), might not strictly identify one of these specific sizes."

We have inserted one sentence immediately before this and one sentence immediately after, and modified the original, to help clarify:

To appropriately interpret the MISR-retrieved size constraint, we refer to the Retrieved Effective Particle Size (REPS), which indicates qualitative changes in the effective size of the retrieved mixture of particle types. The MISR REPS aggregates the contributions of these size components, and the retrieved ANG, which might be more representative of the actual particle size differences (i.e., relative, not absolute), avoids strictly identifying specific sizes. So, for example, an increase in REPS corresponds to a higher AOD fraction of larger components retrieved within in the plume.

17. Line 484 - '... some plumes containing as much as 40% non-spherical...' Perhaps quantify 'some' or change 'some' to 'a very few plumes contain...' as several places (line 480 and line 485) indicate very few plumes have much in the way of non-spherical particles but then 40% is a lot!

Right. We changed "some" to "a few"

18. Line 487 - I assume that the attribution of lower SAE to more non-spherical particles in F plumes is due to the fact that these non-spherical particles are large (re=1.21 um), rather than their non-sphericity?

    That is correct. We have clarified this sentence.

19. Line 623 'there have been no other large scale regional studies' I guess this depends how large scale and regional studies are defined?  For example, Kleinman et al (2020) https://doi.org/10.5194/acp-20-13319-2020 uses in-situ airborne observations to look at particle property changes due to aerosol aging in smoke plumes in the Pacific Northwest region.

    The Kleinman paper certainly does a good job at parsing out the particle chemistry over time. However, the samples were for the most part taken only 2 to 3 hours downwind (whereas we were able to observe smoke up to 10 hours old), and the study included only 9 flights (whereas we observed 663 plumes). Our statement is aimed at emphasizing the contribution the satellite observations can make. For example, we also could study smoke over a wider geographic area.  We have reworded the sentence to say that "…there have been no other studies of such scope relating particle properties to specific aging mechanisms and the timescales over which they occur, made possible by the extensive coverage provided by satellite remote sensing."

**RC2 Comments and Author Responses (AC)**

1.  I am unclear on how actually the plumes are selected from the totality of the MISR dataset. It isn't clearly articulated the criteria. Evidently 50% of the selected plumes are from the MISR plume height archive. Where are the other 50% from? Is the selection done manually? How? Line 230 states "Well-defined plumes…were favored for this analysis." What constitutes well-defined (AOD, something else)? Is it a requirement that it is a smoke plume versus something else? How are you certain you are selecting smoke cases? Are fire anomalies required? I think this needs to be in section 2.2, which rather than providing the information on case selection the title suggests seems more to be about the characteristics of the cases.

    Fair enough. Before running MINX, plumes were manually identified from visible satellite imagery and infrared thermal anomalies. We use NASA's Worldview web app (an open-source website) to look at the MODIS Terra True Color imagery, plus and MODIS Terra hotspots and the Terra orbital track overlayed on top. This allows the user (whether it be the authors of this paper, or others who contribute to the MISR Plume Height Project) to search for smoke plumes within the MISR swath width. It can take practice to identify these plumes and their suitability for the study– usually they are associated with MODIS fire pixels near their source, they are darker than clouds, and appear distinct from the surface below. We do not select plumes where there is a lot of cloud cover obscuring the view of the smoke. We also do not select plumes that we can tell will have very low AOD, as MINX will not be able to retrieve these effectively (again, this takes some practice).

The MISR plume height project contained all possible fires that could be digitized within the MISR swath width for 2017 and 2018 from June through August. Plumes from other years and plumes in May or September of 2017/2018 were identified using Worldview and digitized by the lead author.

We have clarified this in Sections 2.1 and 2.2 to explain how we use the NASA Worldview imagery and MODIS thermal anomalies to identify cases for this study.

2. The main concern I have is that I don't know how to judge the robustness of the retrieval results regarding particle composition presented. My understanding of an algorithm like MISR's RA is that a "best" mixture is determined from its closeness to the observed spectral and angular data. I'm not clear on the orthogonality of the individual components in Table 1, and neither am I clear on how a mixture of these components that minimizes the cost function compares to another mixture that almost minimizes the cost function. Is there degeneracy in the results that admits a different solution? Is it significantly different? Some further presentation of such an error analysis would be important here in order enhance the confidence in the results presented.

We cite the Kahn and Limbacher papers as well as Kahn et al., 2008 and Kahn and Gaitley 2015 to direct readers to discussion of error analysis. Briefly, MISR particle microphysical property information (size, SSA, fraction non-spherical) is qualitative. The algorithm climatology covers the range of properties of major aerosol types, globally, at a precision based on cluster analysis of theoretical sensitivity simulation studies and validated by field experiments to the extent possible. As the references discuss, sensitivity to particle microphysical properties is much more dependent on retrieval conditions (especially AOD) than the retrieved AOD itself, though smoke plumes are especially good targets due to high AOD in general. The retrieval algorithm identifies all mixtures in the climatology that meet sensitivity-based acceptance criteria, and the resulting range of values for each dimension (size, SSA, fraction non-spherical) provides an indication of the information content of that particular retrieval result. We introduce REPS and REPA to capture the qualitative nature of these results, and to emphasize trends in particle size and SSA that are more robust than absolute values.

3. I find the arrows in Figures 8 and 9 not very well justified and suggest they be omitted. If the point is there are clear trends then just plot the line fits with the appropriate statistics, otherwise I think the interpretation is a bit forced.

The arrows present easier visualization of the overall trends, and the scatter of points would make line fits less useful. We take into consideration where deviations from the overall pattern are not statistically significant when comparing adjacent drought categories (for example, comparing D0 directly to D1), whereas the overall trend is statistically significant when taking subsequent drought categories into account (such as comparing D0 to D2, D3 and D4). Also, we reviewed our data and decided on a more conservative interpretation of the trends. As also discussed in our response to Q3 from Reviewer 1, we have therefore

redrawn the arrows and made the appropriate changes in the text. We also better clarify the justification for the arrows in both figure captions.

4. Line 54-55: I think a comment here on pyroCb would be useful to add, to acknowledge the growing interest in this type of fire.

   Ok. We have added the following: "Under certain meteorological conditions, smoke plumes can even form PyroCumulonimbus, propelling smoke into the upper troposphere or lower stratosphere (e.g., Peterson et al., 2017); such events are relatively rare and are beyond the scope of the current study, but are possibly becoming more frequent."
   The reference is: Peterson, D. A., Hyer, E. J., Campbell, J. R., Solbrig, J. E. & Fromm, M. D., 2017. A conceptual model for development of intense pyrocumulonimbus in western North America. *Mon. Weather Rev. 145, 2235–2255*, doi:10.1175/MWR-D-16-0232.1.

5. Table 1: For the utility of modelers, please include in the supplement further information on the particle properties summarized here. For example, the mode radius and width and the refractive indices at the MISR channels.

   Done. However, we note that although any retrieval algorithm can generate "numbers," the particle size and refractive index results are actually qualitative, given the limitations of the remote sensing technique. We have made a considerable effort to convey this to readers here and in previously published work, to reduce the likelihood of over-interpretation, e.g., by emphasizing REPS and REPA instead of specific sizes and light-absorption properties.

6. Line 91-96: This construct is unclear. Do you mean in the end that you consider new particle formation and condensation/hygroscopic growth as the distinct aging mechanisms?

   That is correct. We have clarified this last sentence.

7. Line 103: "Most current transport and climate models…" I think it is more accurate to say that most current climate models do not at present incorporate brown carbon at all. The treatment of smoke as being a mixture of black carbon, organic (white) carbon, and sulfates leads to I think what you mean by "BlS" in this paper.

   Yes, most models do not define a brown carbon component, and many treat all absorbing particles as "black carbon." This sentence allows us to use the new terminology rather than reverting to "brown carbon" and "black carbon."

8. Line 234: You mean the lower 48 here, as Alaska is part of your domain and is part of the US.

   Correct, we have fixed this.

9. Line 293: 0.667 degrees is inconsistent with 0.625 degrees a few lines earlier. I believe the MERRA-2 fields are uniformly available on the 0.625 x 0.5 degree grid.

Correct, we have fixed this.

10. Line 327: "event" not "even"

Fixed.

11. Line 449: should be "compared to G"

Fixed.

12. Line 449/450: "More on particle properties in subsequent sections" seems a bit informal. Maybe "More information on particle properties is presented in subsequent sections."

Agreed, we have rephrased based on your suggestion.

13. Line 461: Add pointer to Table 4 in this first sentence.

Done.

14. Line 467: Suggest starting a new paragraph with "To help interpret ANG…"

Agreed, done.

15. Line 476: Add reference to Figure 3 in this sentence that begins "As expected…"

We cite this figure in the next sentence, which also requires a pointer to Figure 3.

16. Line 480: Add reference to Table 4 here again.

Done.

17. Line 587: I think "BlS" is meant instead of "BrS"

Correct, this has been fixed.